# Probing the molecular determinants of Ty1 retrotransposon restriction specificity in yeast

Sean L. Beckwith[1,2☯¤a], Matthew A. Cottee[3☯¤b], J. Adam Hannon-Hatfield[1¤c], Abigail C. Newman[1], Emma C. Walker[1], Justin R. Romero[2], Jonathan P. Stoye[4], Ian A. Taylor[3*], David J. Garfinkel[1*]

1 Department of Biochemistry and Molecular Biology, University of Georgia, Athens, Georgia, United States of America, 2 Department of Biology, Hope College, Holland, Michigan, United States of America, 3 Macromolecular Structure Laboratory, The Francis Crick Institute, London, United Kingdom, 4 Retrovirus-Host Interactions Laboratory. The Francis Crick Institute, London, United Kingdom

☯ These authors contributed equally to this work
¤a Current address: Department of Biology, Hope College, Holland, Michigan, United States of America
¤b Current address: AstraZeneca, Discovery Centre, Cambridge, United Kingdom
¤c Current address: Department of Biochemistry, Emory School of Medicine, Emory University, Atlanta, Georgia, United States of America
* ian.taylor@crick.ac.uk (IAT); djgarf@uga.edu (DJG)

## Abstract

The evolutionary history of retrotransposons and their hosts shapes the dynamics of transposition and restriction. The *Pseudoviridae* of yeast includes multiple Ty1 LTR-retrotransposon subfamilies. *Saccharomyces cerevisiae* prevents uncontrolled retrotransposition of Ty1 subfamilies using distinct mechanisms: canonical Ty1 is inhibited by a self-encoded restriction factor, p22/p18, whereas Ty1' is inhibited by an endogenized restriction factor, Drt2. The minimal inhibitory fragment of both restriction factors (p18m and Drt2m) is a conserved C-terminal capsid domain. Here, we use biophysical and genetic approaches to demonstrate that p18m and Drt2m are highly specific to their subfamilies. Although the crystal structures of p18m and Drt2m are similar, three divergent residues found in a conserved hydrophobic interface direct restriction specificity. By mutating these three residues, we re-target each restriction factor to the opposite transposon. Our work highlights how a common lattice-poisoning mechanism of restriction evolved from independent evolutionary trajectories in closely related retrotransposon subfamilies. These data raise the possibility that similar capsid-capsid interactions may exist in other transposons/viruses and that highly specific inhibitors could be engineered to target capsid interfaces.

## Author summary

Transposable elements comprise a large fraction of many genomes and occupy approximately fifty times more sequence space in the human genome than protein-coding sequences. Retrotransposons, which have life cycles similar to

**Data availability statement:** The atomic coordinates and structure factors for Ty1′ CA-CTD, Ty1′ CA-CTD (F323S), and Drt2m(SS) have been deposited in the Protein Data Bank under accession numbers 9RXW, 9RXX, and 9RXY, respectively. Whole blot images underlying Figs 5, 6 and S2 are provided in S1 Raw Images. Source transposition data for Figs 1, 5 and S2; sedimentation data for Fig 3; MALLS data for Fig 4 and sequence data for S1 Fig are contained within S1 Data File. All remaining data are contained within the article and supporting information.

**Funding:** Funding Statement This work was supported by the Francis Crick Institute, which receives its core funding for IAT from Cancer Research UK (CC2029), the UK Medical Research Council (CC2029) and the Wellcome Trust (CC2029). This work was also funded by NIH grants to DJG (R01GM124216 and R01GM156837) and an NIH Postdoctoral Fellowship to SLB (F32GM139247). The funders had no role in study design, data collection and analysis, decision to publish, or preparation of the manuscript.

**Competing interests:** The authors have declared that no competing interests exist.

retroviruses, use a "copy and paste" mechanism that increases their copy number with each new insertion. We previously described two distinct mechanisms in the budding yeast *Saccharomyces* by which two subfamilies of the Ty1 retrotransposon are regulated to prevent their copy number from increasing. Both subfamilies are inhibited by a small protein derived from the Ty1 capsid, although the source of this inhibitory protein differs. The inhibitor interferes with the assembly of virus-like particles in the cell where the retrotransposon carries out enzymatic reactions required for its replication. In this study, we show that the inhibitory proteins have remarkably similar structures and yet are highly specific to their respective retrotransposons. We uncover three amino acids where the two inhibitors differ and, by mutating these residues, are able to re-target them to the opposite retrotransposon. This work raises the possibility that highly specific inhibitors may have evolved, or could be engineered, to restrict other transposable elements or viruses.

## Introduction

Retrotransposons replicate through an RNA-intermediate, increasing copy number with each transposition event [1,2]. Transposons are powerful forces in genome evolution and uncontrolled replication can be deleterious to the host genome [3–5]. Therefore, a variety of defense mechanisms have evolved to restrict the mobility of transposons, including RNAi pathways, and SAMHD1, APOBEC, and MOV10 restriction factors [6]. The budding yeast *Saccharomyces cerevisiae* and closely related species such as *S. paradoxus* lack these defense mechanisms, yet transposons comprise only a small fraction of the yeast genome [7]. *S. cerevisiae* harbors members of the retrovirus-like *Pseudoviridae* (Ty1, Ty2, Ty4, Ty5) and *Metaviridae* (Ty3) families of long-terminal repeat (LTR)-retrotransposons [8–10]. Ty1 is the most abundant retrotransposon in many strains, with about 32 full-length copies present in the reference strain S228C [10–12]. Ty1 contains two open reading frames and encodes Gag and Pol proteins. Ty1 Gag is equivalent to retroviral Gag and provides both capsid (CA) and nucleocapsid (NC) nucleic acid chaperone functions. Ty1 Pol is a polyprotein that is proteolytically processed into protease (PR), integrase (IN), and reverse transcriptase (RT), enzymes, each of which is required for retrotransposition [11].

Through its capsid function, Ty1 Gag assembles virus-like particles (VLPs) that serve as cytoplasmic reaction vessels where Ty1 protein maturation occurs and Ty1 mRNA is reverse transcribed [1,13–16]. Unlike HIV-1 reverse transcription which is completed within the nucleus [17,18], available biochemical and genetic evidence suggests that Ty1 reverse transcription occurs in cytoplasmic VLPs [19–22]. A pre-integration complex minimally containing Ty1-cDNA and IN is imported into the nucleus via a nuclear localization sequence present on IN and integrated predominately at loci upstream of RNA polymerase III-transcribed genes [23–25]. Structural studies of retroviruses, Ty3 Gag, and the domesticated transposon capsid ARC reveal how CA forms hexameric and pentameric building blocks that assemble into fullerene VLPs and viral cores [26–29]. Ty1 Gag contains N- and C-terminal CA domains (NTD/CTD) that direct assembly of CA into Ty1 VLPs [30].

PLOS Genetics

Uncontrolled Ty1 retrotransposition is prevented by a mechanism termed copy number control (CNC) [31]. This form of transposon control is mediated by a self-encoded restriction factor, p22, derived from the *GAG* coding sequence. p22 is synthesized from a transcript initiated within *GAG*, and p22 and its PR-catalyzed product p18 associate with Gag during VLP assembly [32,33]. The minimal restriction factor, p18m, is identical to the CTD of Ty1 Gag [30]. Structural and genetic analyses of p18m reveal that p18m and Ty1 Gag interact through a hydrophobic region that is shared between both proteins, termed the Dimer-1 interface [30]. These observations suggest a lattice-poisoning model of restriction by p18m in which the CTD binds full-length Gag and forms dead-end intermediates in VLP assembly.

Our understanding of yeast Ty1 comes from studies of canonical Ty1 (Ty1c), but *S. cerevisiae* also contains Ty1' elements, a subfamily of Ty1 LTR-retrotransposons [7,10,12]. Ty1' is not subject to the canonical CNC mechanism and is instead restricted by *DRT2*, a gene present at a fixed chromosomal locus that encodes the Ty1' Gag CTD equivalent to p22 [34]. Drt2 also associates with Ty1' Gag while forming VLPs and likely uses a lattice-poisoning mechanism to restrict retrotransposition. This form of retroelement control is conceptually similar to the mouse Fv1 restriction factor which encodes CA-derived proteins that can inhibit murine leukemia retroviruses (MLV); furthermore, different alleles of Fv1 restrict different MLV variants [35].

Here, we compare the similarities and differences of p22 and Drt2 mediated restriction. We investigate cross-restriction between the Ty1c and Ty1' transposon control mechanisms and find that p18m and Drt2 display subfamily-specific restriction. We determined crystal structures of the Ty1' CA-CTD and a minimal fragment of Drt2 (Drt2m) and use biophysical and genetic analyses to reveal how divergent residues in Dimer-1 confer restriction specificity. Our work details two highly specific restriction factors in closely related transposon subfamilies and provides insights from nature into engineering precisely targeted inhibitors using CA-CTDs for viruses and transposons.

## Results

### Restriction mechanisms are specific to Ty1 subfamilies

Retrotransposition of the Ty1c and Ty1' subfamilies is controlled by distinct restriction mechanisms. Ty1c is under CNC via the self-encoded p22 restriction factor [30,32,33,36]. In contrast, Ty1' is inhibited by an endogenized Ty1' gene fragment encoding a p22-like protein, Drt2, that is present at a fixed chromosomal locus in diverse strains but absent in the S288C reference strain [34] (Fig 1A). We sought to determine if these distinct restriction mechanisms could cross-restrict across the closely related Ty1 subfamily members or if they are specific to their respective transposon. We tested Ty1c and Ty1' retrotransposition using a plasmid-borne element marked with the *his3-AI* retrotranscript indicator gene [37] (Fig 1B). Integration of Ty1 cDNA either by retrotransposition or recombination, collectively termed retromobility, is measured using *his3-AI* by monitoring for histidine prototrophy, which requires the transposon to have successfully completed RNA splicing, reverse transcription, and integration. Retromobility was measured in isogenic *S. cerevisiae* strains containing the endogenous restriction factor (*DRT2*) or a chromosomal deletion (*drt2Δ*) but lacking any full-length Ty1c or Ty1' elements [34]. Ty1' retromobility is inhibited four-fold by the presence of *DRT2* as previously reported [34], but Ty1c retromobility showed no significant change in the presence or absence of *DRT2* (Fig 1C & S1 Table). These data suggest that Drt2 restriction is specific to Ty1', although it is possible that low endogenous expression levels may limit the effect of Drt2 restriction [34]. Given the high amino acid sequence similarity between Ty1c and Ty1' (91.3% identity in Gag-Pol and 78.9% identity in Gag), we sought to understand the molecular determinants responsible for subfamily specific restriction. We focused further analyses on cross-restriction between p18m or Drt2m (the equivalent residues in Drt2, Gag coordinates 259–355) with Ty1c or Ty1' to avoid confounding effects of potentially forming mixed VLPs that contain both Ty1c and Ty1' Gag proteins (which may also impair retrotransposition) and to allow direct comparisons between restriction factor structure and function.

### Subfamily specific residues in the Dimer-1 interface

We analyzed the amino acid variability of p18m, Drt2m, and the equivalent region of Ty1' Gag in 148 elements (see Methods) and found that each protein is highly conserved (S1 Fig): p18m is 99.0% identical across 98 sequences,

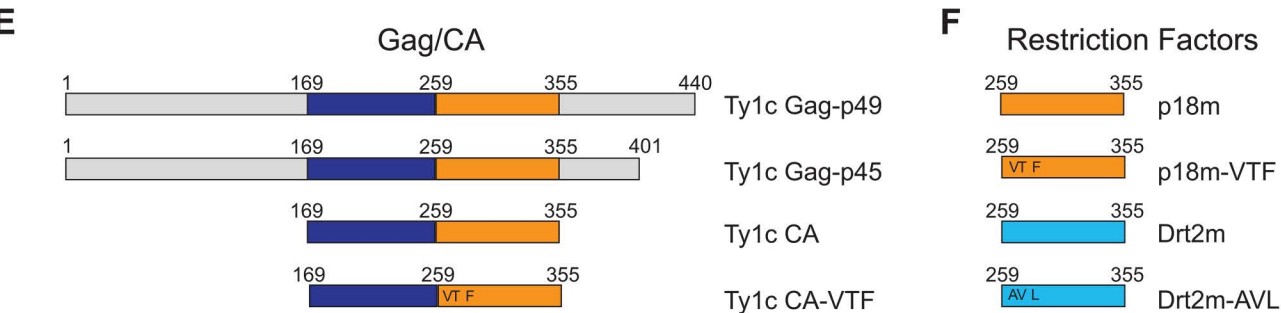

**Fig 1. Subfamily specific restriction of Ty1 elements. (A)** Schematic contrasting the two restriction mechanisms evolved for Ty1c and Ty1': the self-encoded p22 protein restricts Ty1c whereas an endogenized gene at a fixed locus, *DRT2*, expresses an inhibitory protein against Ty1'. **(B)** Schematic illustrating the transposon expression plasmid and the chromosomal *DRT2* restriction factor. pTy1c*his3-AI* is on a *URA3* marked vector and pTy1'*his3-AI* is on a *TRP1* marked vector. Each transposon is expressed under its native promoter and contains the *his3-AI* retromobility indicator gene; histidine prototrophy requires retromobility. **(C)** Quantitative mobility assay. Each bar represents the mean of the four independent measurements displayed as points. The error bar center represents the mean of the four measurements and the error bar extent ± the standard deviation. Significance is

calculated from a two-sided Student's *t*-test compared with wildtype (n.s not significant, *** $p < 0.001$. Exact *p*-values are provided in S1 Table). Significant fold-change in restriction compared to wildtype is indicated above the bars. **(D)** Sequence alignment of p18m and Drt2m restriction factors. Numbering is according to the equivalent position in Ty1c Gag. Positions of α-helices observed in crystal structures are indicated by the green bars above the alignment. Divergent residues are indicated by boxes; thick red boxes indicate Dimer-1 interface residues where p18m differs from Drt2m. Gray shaded residues indicate residues where Dimer-2 suppression serine mutations have been made. **(E)** Bar representation of Gag and CA, residue numbers correspond to the Gag sequence. The CA-NTD is colored dark blue and CA-CTD is orange. **(F)** p18-type restriction factors, residue numbers correspond to the Gag sequence. p18m is colored orange like Ty1c CA-CTD and Drt2m is light blue. **(E & F)** VTF and AVL indicate the tri-point mutations that convert the Ty1c CA or p18m to a Drt2m interface (AVL to VTF) and the Drt2m to a Ty1c interface (VTF to AVL).

Drt2m is 91.8% identical across 15 sequences, and the p18m-equivalent region of Ty1' Gag is 92.8% identical across 35 sequences. We aligned representative sequences of p18m and Drt2m and identified 17 amino acid differences (Fig 1D). Three divergent residues fall within Dimer-1 of the p18m crystal structure: A266, V270, and L312 (hereafter AVL). In both Drt2m and Ty1' Gag these positions are instead V266, T270, and F312 (hereafter VTF). Importantly, all three residues were 100% conserved as either AVL or VTF throughout the 148 elements examined (S1 Fig). Given the importance of Dimer-1 in p18m restriction, we hypothesized that these three residues may be the determinants of subfamily restriction specificity. Therefore, we probed the functional role of these residues using biophysical and genetic approaches with wildtype and mutant constructs of Gag or Ty1c CA and p18m and Drt2m restriction factors (Fig 1E-F).

## Structure of Ty1' CA-CTD and the Drt2m restriction factor

We reported the 2.8 Å crystal structure of p18m, which revealed an extensive hydrophobic interface, Dimer-1, that forms a homotypic CA-CTD interface with Ty1c Gag to inhibit VLP assembly [30]. Given that Ty1' CA and Drt2 also encode p18-like domains and to understand if they had the same Dimer-1 properties as p18m, we expressed Drt2m and the equivalent residues in Ty1' CA-CTD (Gag coordinates M259-Q351) and determined their crystal structures. We observed p18m higher order aggregation through a weaker Dimer-2 interface [30] that was ameliorated by introduction of a Dimer-2 suppressing mutant (F323S). Therefore, we included F323S in Ty1' CA-CTD constructs and in Drt2m F323S/Y329S (SS) for crystallographic analyses and F323S/Y326S/Y329S (SSS) for biophysical analyses. Inclusion of serine substitutions reduced the aggregation tendency and improved solubility necessary for structural studies. To validate the use of these aggregation suppressing mutations, we verified that Drt2m(SSS) is stably expressed in yeast (S2A Fig). Furthermore, Drt2m(SSS) and p18m(F323S) maintained robust restriction activity in yeast transposition assays (S2B Fig). Dimer-2 mutations decrease restriction activity by less than an order of magnitude, that is not significant in p18m(F323S) and is marginally significant in Drt2m(SSS). Together, these results suggest that Dimer-2 is dispensable for restriction (S2B Fig). However, Dimer-2 does contribute to Gag function as Ty1c Gag(F323S) lowers retromobility approximately two orders of magnitude and is highly significant (S2C Fig). This may be due to Dimer-2 interactions promoting oligomerization required for Gag capsomers to assemble the VLP [30].

Using a p18m monomer (PDB: 7NLH) polyalanine search model, we determined structures by molecular replacement at 1.60 and 1.62 Å resolution for wildtype and Ty1' CA-CTD (F323S), respectively, and at 3.18 Å for Drt2m(SS). Details of the data collection structure determination and refinement are included (S2 Table). Inspection of the structures reveals all helical domains with arrangements and packing resembling that of Dimer-1 observed in p18m [30] and confirm that the F323S mutation does not alter interactions at the Dimer-1 interface. The asymmetric unit of wildtype and Ty1' CA-CTD (F323S) crystals both contain two monomers arranged through a single Dimer-1 interface and the Drt2m(SS) crystals contain four Drt2m(SS) monomers arranged as two dimer-pairs through Dimer-1 interfaces (S3A Fig). 3D comparison of the Ty1' CA-CTD (F323S) and Drt2m(SS) dimers also reveals the strong similarity (S3B Fig) with RMSD of 0.5 Å over 160 Cα backbone atoms. Further comparison of the p18m (Fig 2A) dimer with that of Drt2m(SS) (Fig 2B) reveals the strong similarity demonstrated by the 3D superposition (Fig 2C) of RMSD of 0.7 Å over 155 Cα backbone atoms.

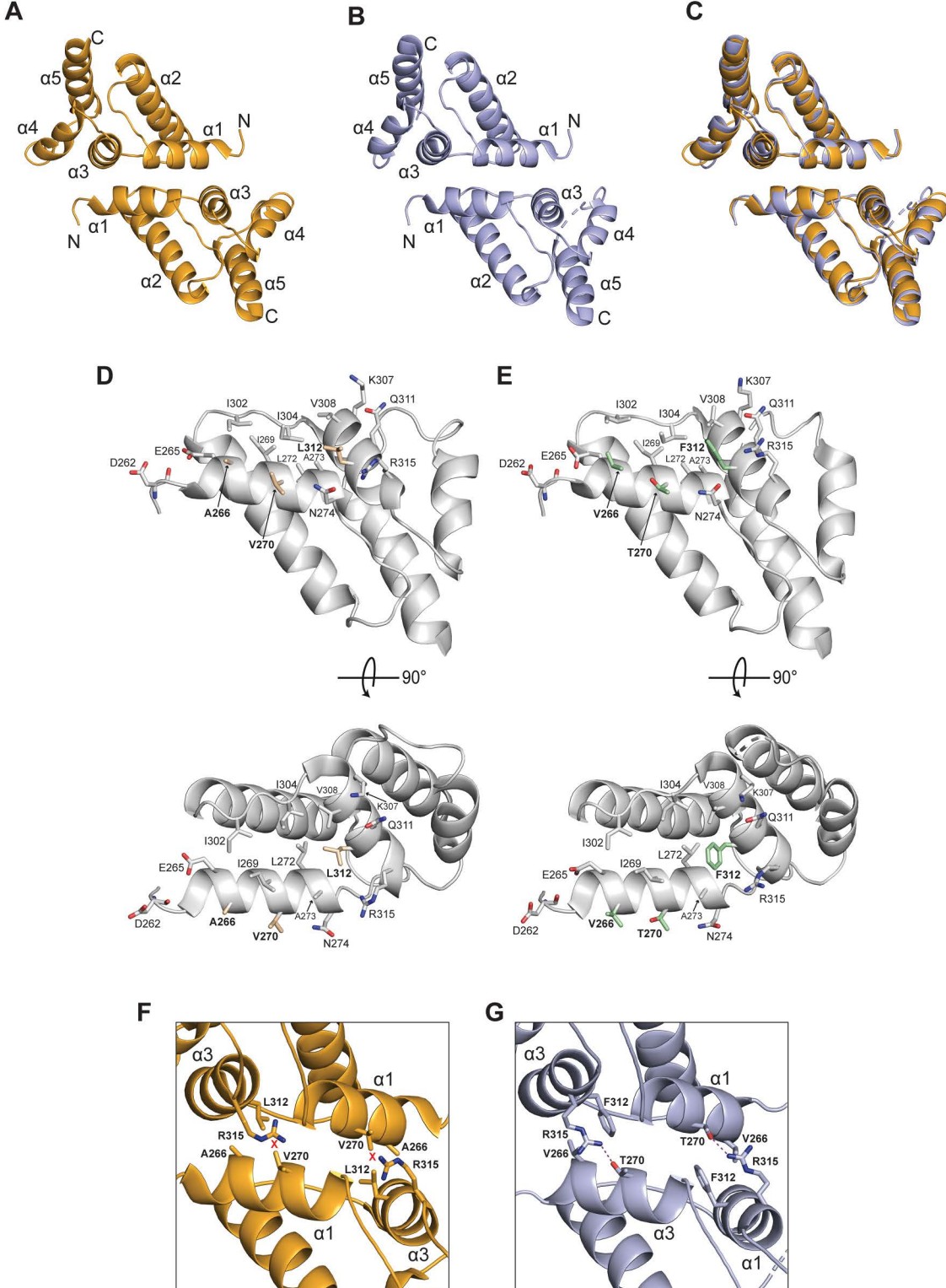

**Fig 2. Crystal structures of p18m and Drt2m restriction factors.** Crystal structures of (**A**) p18m and (**B**) Drt2m(SS) dimers. The protein backbone is shown in cartoon representation, p18m in orange and Drt2m in pale blue. Helical secondary structures in each monomer are labelled sequentially from N- to C-terminus. (**C**) 3D structural superposition of p18m and Drt2m dimers. Coloring and orientation are the same as in **A** and **B**. Structures were aligned using 155 backbone Cα atoms, yielding an RMSD of 0.7 Å. (**D**) p18m and (**E**) Drt2m Dimer-1 interfaces. The view is perpendicular to (upper) and

into (lower) the interface of a single monomer. The protein mainchain is shown in gray cartoon. Residues that make apolar and salt bridge interactions at the interface are shown in stick representation. Specificity determining residues in p18m (A266, V270 and L312) and Drt2m (V266, T270 and F312) are highlighted with bold labelling and shown in light brown and green respectively. **(F & G)** View of the Dimer-1 interface of **(F)** p18m and **(G)** Drt2m. View and cartoon coloring is the same is **A** and **B**. The variable interfacial residues (A/V)266, (V/T)270 and (L/F)312 together with R315 on helices α1 and α3 are shown in stick representation. The T270-R315 hydrogen bonding interaction at the Drt2m interface is indicate by the purple dashes. Red crosses indicate a lack of hydrogen bonding in p18m interface.

Taken together, these structures show that p18m, Ty1' CA-CTD and Drt2m collectively share the same Dimer-1 interface responsible for restriction in the Ty1c system. All residues forming this interface are conserved apart from determinant residues 266, 270 and 312 that are AVL in Ty1c Gag and p18m (Fig 1D) and VTF in both Ty1' Gag and Drt2m (Figs 1D & S3C). Inspection of the location of these divergent residues reveals that in p18m (Fig 2D) and Drt2m (Fig 2E) the three divergent residues lie at the center of the hydrophobic Dimer-1 interface. In p18m, within this interface the sidechain of A266 extends from helix α1 of Monomer-A and packs into a complementary pocket on Monomer-B, lined by L312 sidechain extending from helix α3 (Fig 2F). This close packing results in a distance between the A266 and L312 Cα atoms of 6.2 Å. By contrast, in Drt2m and Ty1' CA-CTD, it is the V266 sidechain on helix α1 that closely packs against F312 on helix α3 and because of the greater steric bulk of the V and F residues the distance between the V266 and F312 Cα atoms is increased to 6.8 Å (Fig 2G). Concurrent with the increased spacing in the VTF interface of Drt2m and Ty1' CA-CTD, the side chain of the conserved residue R315 can adopt a rotamer that allows it to swing in and hydrogen bond with T270 which is not possible with V270 in p18m. Thus, this "arginine-flip" mechanism provides a further level of discrimination and specificity between AVL and VTF Dimer-1 interfaces.

These structural data show how the residue configuration, their interaction networks, and the complementary surfaces they create at the Dimer-1 interface are subtly different between the Ty1c and Ty1' subfamilies. As Dimer-1 is responsible for restriction by p18m and Drt2m, the AVL/VTF subtypes conserved within each family may limit restriction specificity to within each family as, although still capable of hydrophobic interactions, the incompatibility of the heterologous pairings abolishes or severely diminishes the capacity for cross-family restriction of Ty1 elements.

## Solution oligomerization properties of restriction factors

Given our structural data showing the strong similarity of Drt2m to p18m and that p18m forms stable dimers in solution [30], sedimentation equilibrium analytical ultracentrifugation (SE-AUC) was employed to compare the solution oligomerization properties of Drt2m(SSS) and p18m(F323S). Multispeed SE-AUC studies were carried out at varying protein concentration (20–90 µM). Typical equilibrium distributions were recorded at the three speeds using both absorbance and interference optics (Fig 3). Protein and solution hydrodynamic parameters along with results of data fitting are presented in S3 Table. Analysis of individual gradient profiles using an ideal individual species model showed poor fitting and molecular weight concentration dependencies ranging from 27.0 to 32.0 kDa for Drt2m(SSS) and 34.4 to 37.9 kDa for p18m(F323S). These observations suggest that even with the Dimer-2 suppressing mutations there was a tendency for self-association to greater than the dimer molecular mass (~23 kDa). Moreover, and as might be expected, the tendency for further self-association was more apparent in p18m(F323S) than in Drt2m(SSS). Given these observations, the data were then fitted globally using a monomer-dimer-tetramer model (Fig 3 & S3 Table). The application of this model improved fits comprising associating monomer-dimer equilibria with $K_D^{(1-2)}$ of 0.71 µM and 0.48 µM for p18m(F323S) and Drt2m(SSS), respectively, together with further weakly associating dimer-tetramer equilibria, $K_D^{(2-4)}$ of 30.5 µM for p18m(F323S) and 277 µM for Drt2m(SSS). Therefore, p18m(F323S) and Drt2m(SSS) form dimers in solution with sub-micromolar $K_D$ with some tendency for further p18m(F323S) self-association at higher concentration but that in Drt2m(SSS) is largely suppressed by the introduction of the triple serine mutation.

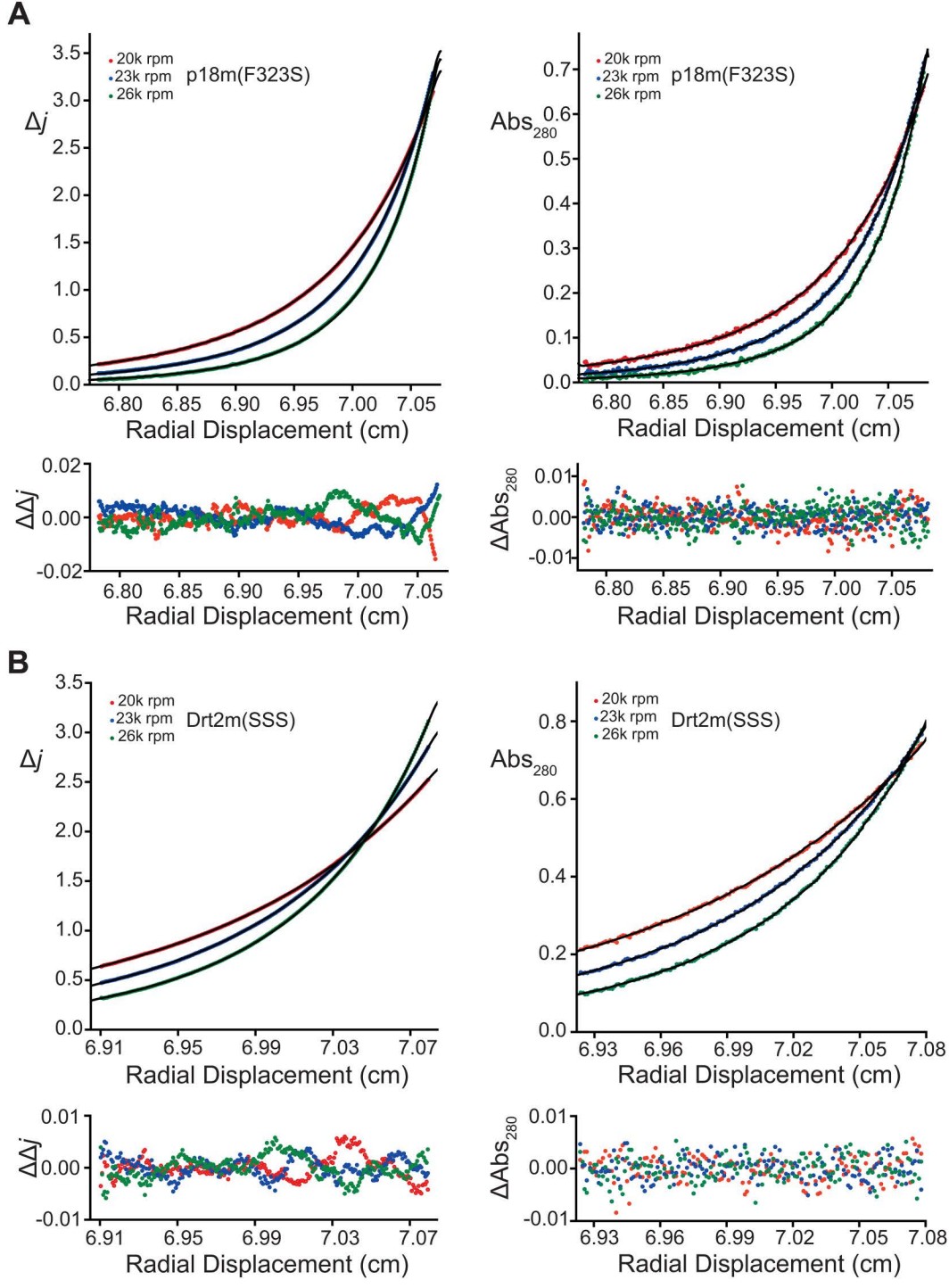

**Fig 3. Sedimentation equilibrium analysis of p18m and Drt2m self-association. (A** and **B)** Upper panels are the multi-speed sedimentation equilibrium profiles determined from interference (left) and absorbance (right) data collected on **(A)** p18m(F323S) at 45 µM and **(B)** Drt2m(SSS) at 44 µM. Data was recorded at the speeds indicated. Data points in each panel are colored according to the speed. The solid lines represent the global best fit to the data using a monomer-dimer-tetramer model (p18m(F323S); $K_D^{(1-2)}$ = 0.71 µM, $K_D^{(2-4)}$ = 30.5 µM, reduced $\chi^2$ = 0.407) and (Drt2m(SSS); $K_D^{(1-2)}$ = 0.48 µM, $K_D^{(2-4)}$ = 277 µM; reduced $\chi^2$ = 0.190). The lower panels are the residuals to the fit, the color of points corresponds to that of the fitted profile in the upper panels, see also S3 Table.

## SEC-MALLS analysis of capsid-restriction factor exchange

To probe the mechanism and specificity of p18m- and Drt2m-mediated restriction, we wanted to assess whether it would be possible to swap specificity between systems by replacing AVL with VTF or VTF with AVL at Dimer-1 interfaces. Size exclusion chromatography coupled laser light scattering (SEC-MALLS) can be used to determine absolute and average molar mass of proteins by combining the scattered light intensity and protein concentration measurements of samples eluting within chromatographic peaks. Therefore, we performed SEC-MALLS to assess the capacity for restriction factors p18m(F323S) and Drt2m(SSS) along with Dimer-1 swapped variants p18m-VTF(F323S) and Drt2m-AVL(SSS) (Fig 1E-F) to exchange subunits with CA of canonical Ty1c CA(F323S) and the Dimer-1 swapped derivative Ty1c CA-VTF(F323S). All constructs contained either the F323S or the SSS Dimer-2 mutations to suppress higher order aggregation that would interfere with the chromatographic analysis. SEC-MALLS initially carried out with individual components over protein concentrations ranging from 12.5-200 μM demonstrated that Ty1c CA(F323S) and Ty1c CA-VTF(F323S) as well as p18m(F323S), Drt2m(SSS), p18m-VTF(F323S) and Drt2m-AVL(SSS) restriction factors all formed stable dimeric species (S4 Fig), which is consistent with those observed for p18m(F323S) and Drt2m(SSS) by sedimentation equilibrium (Fig 3). We next tested the capacity for each of the four restriction factors to exchange subunits with Ty1c CA dimers by assessing the degree of heterodimer formation that occurred upon 20-hour incubation. These data (Figs 4 & S5) reveal that, depending on the combination, different degrees of exchange occur. For the canonical pairing of p18m and Ty1c CA, a large proportion of heterodimer is produced, observed as an intermediate peak that elutes with a retention time between that of the Ty1c CA dimer and the p18m dimer (Fig 4A, red chromatogram). Quantification of the amount of exchanged material, as the fraction sum residual ($\theta_{res}$), determined by subtraction of a calculated Ty1c CA - p18m sum-chromatogram from the experimental chromatogram output from the differential refractometer (S5 Fig) yields a measure for the degree of exchange (Table 1). For the Ty1c CA(F323S) - p18m(F323S) pair, a value of 25.4% is obtained demonstrating the capacity for p18m and Ty1c CA homodimers to readily exchange subunits with one another. Analysis of the other pairings reveals that when the Dimer-1 interfaces are mismatched, AVL in one and VTF in the other (Fig 4C, 4D, 4G & 4H), then there is no appearance of an intermediate peak and the $\theta_{res}$ quantitation yields values at around 2.2% that we interpret as a background for no exchange. By contrast, other pairs with matched Dimer-1 interfaces, AVL with AVL or VTF with VTF (Fig 4B, 4E & 4F), show varying degrees of exchange that ranges from strong, 17.6%, exhibited by the Ty1c CA-VTF(F323S) - p18m-VTF(F323S) pair to intermediate values, 6.2% for Ty1c CA(F323S) - Drt2m-AVL(SSS) and a weaker 3.2%, albeit close to background, for Ty1c CA-VTF(F323S) - Drt2m(SSS) pairings (Table 1).

Our data clearly demonstrate the *in vitro* capacity for restriction factors to exchange subunits with Ty1 CA dimers through swapping at the Dimer-1 interface further supporting the lattice-poisoning model for p18m and Drt2m restriction of Ty1c and Ty1' through inhibition of VLP assembly [30,34].

## Cross-restriction in vivo

Our structural and biophysical analyses now indicate Dimer-1 compatibility as a major determinant of restriction specificity. To establish the role of the subfamily specific Dimer-1 residues *in vivo*, we employed an approach previously used to assess p18m restriction of Ty1c in a *S. paradoxus* yeast strain lacking any full-length Ty elements [30]. An ectopic overexpression system comprising a *his3-AI* marked Ty1c or Ty1' element on a low-copy plasmid and the p18m or Drt2m restriction factor on a multi-copy plasmid was utilized for co-expression in yeast (Fig 5A). Strains contained either full length Ty1c or Ty1' elements with wildtype Gag sequences, or Ty1c-VTF (converting the Dimer-1 interface to the Ty1'/Drt2 residues), or Ty1'-AVL (converting the Dimer-1 interface to the Ty1c/p18 residues). In addition, strains also contained wildtype p18m or Drt2m restriction factors, or p18m-VTF, or Drt2m-AVL. Strains were generated containing all pairwise combinations of transposon and restriction factor as well as an empty vector lacking the restriction factor in control strains. All the constructs expressed well with slight variability in protein levels (Fig 5B). We observed a striking effect of mutating the Dimer-1 residues on retromobility restriction (Fig 5C & S1 Table). p18m strongly restricts Ty1c but p18m-VTF was

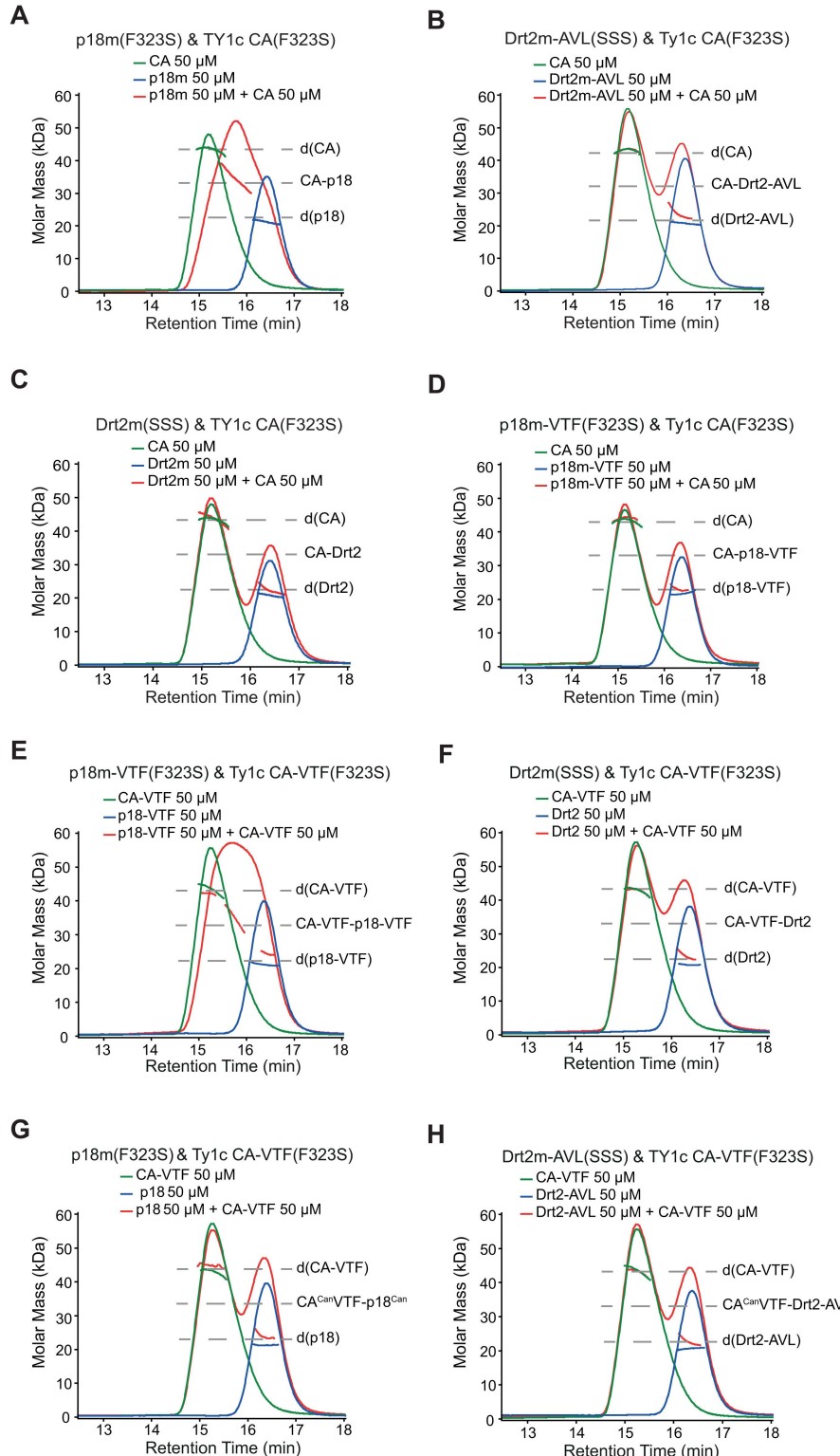

**Fig 4. SEC-MALLS analysis of restriction factor specificity.** SEC-MALLS analysis of **(A-D)** Ty1c CA(F323S) and **(E-H)** Ty1c CA-VTF(F323S) interaction with p18m(F323S), p18m-VTF(F323S), Drt2m(SSS) and Drt2m-AVL(SSS) restriction factors. In each panel, the differential refractive index (dRI) is plotted against column retention time for sample loadings of 50 µM Ty1c CA(F323S) or Ty1c CA-VTF(F323S) (green), 50 µM p18m(F323S),

p18m-VTF(F323S), Drt2m(SSS) or Drt2m-AVL(SSS) (blue) and 50 µM equimolar mixtures (red) as indicated. The molar mass, determined at 1-second intervals throughout the elution of peaks, is plotted as points. The molecular masses of p18m or Drt2m homodimers, p18m-Ty1c CA or Drt2m-Ty1c CA heterodimers and the Ty1c CA or Ty1c CA-VTF homodimers are indicated with the gray dashed lines.

**Table 1. CA-specificity of p18m restriction factor exchange.**

| CA/p18 | p18m (F323S) | Drt2m (SSS) | p18m-VTF (F323S) | Drt2m-AVL (SSS) |
|---|---|---|---|---|
| TY1c CA(F323S) | *25.4 | 2.7 | 2.0 | 6.2 |
| TY1c CA-VTF(F323S) | 2.0 | 3.2 | 17.6 | 2.3 |

*The sum fraction residual θres presented as percentage. The values from pairs greater than 2x the average background measurement (2.2±0.3%) are underlined. Non-compatible non-exchanging combinations are shown in grey.

| CA/p18 | p18m (F323S) | Drt2m (SSS) | p18m-VTF (F323S) | Drt2m-AVL (SSS) |
|---|---|---|---|---|
| TY1c CA(F323S) | 3* | 0 | 0 | 2 |
| Ty1c CA-VTF(F323S) | 0 | 1 | 3 | 0 |

*Degree of exchange observed by MALLS: 3, Largely exchanged > 5x background; 2, appreciable exchange > 2x background; 1, weak exchange > 1.3x background, 0, non-observable < 1.3x background.

less potent with restriction reduced by over a thousand-fold. Conversely, Drt2m had extremely weak restriction against Ty1c (1.6-fold restriction, $p = 0.026$) but Drt2m-AVL displayed a several thousand-fold stronger restriction. The same effect was detected when mutating the Gag residues instead of the restriction factor. Ty1c-VTF largely escapes restriction by p18m but is strongly restricted by p18m-VTF. Drt2m strongly restricts Ty1c-VTF unlike wildtype Ty1c, but this restriction is completely abolished by mismatching the Dimer-1 interface with Drt2m-AVL. The same trend was observed in Ty1' albeit to a milder degree. Ty1' is a less active element than Ty1c in this system and is restricted in a smaller dynamic range. Nonetheless, all pairwise tests against Ty1' and Ty1'-AVL also demonstrated significantly stronger restriction in homotypic Dimer-1 interfaces and weaker restriction in mismatched heterotypic Dimer-1 interfaces.

### Restriction factor-mediated disruption of VLP assembly

Both p18m and Drt2m associate with VLPs and likely restrict through a blocking-of-assembly mechanism using the Dimer-1 interface [30,34]. To determine if restriction factor specificity also affects VLP assembly, we analyzed sedimentation profiles of VLPs through a 7–47% continuous sucrose gradient in the co-expression strains used to determine retromobility (Fig 6). Additional controls verified that wildtype Ty1c forms VLPs that accumulate in more dense fractions near the bottom of the gradient (peak Gag fractions indicated with bar) while restriction factors p18m and Drt2m expressed alone accumulate in less dense fractions near the top of the gradient. p18m disrupts Ty1c VLP assembly and peak Gag fractions are markedly redistributed towards the top of the gradient. However, this Gag redistribution is not observed in the presence of p18m-VTF despite a minor fraction of p18m-VTF appearing in higher density fractions like wildtype p18m; instead, only a slight broadening of peak Gag fractions is observed. In the presence of Drt2m, peak Gag fractions still accumulate near the bottom of the gradient, but the presence of Drt2m-AVL noticeably shifts peak Gag fractions towards less dense fractions at the center of the gradient. These results suggest that the subfamily specific VTF/AVL Dimer-1 residues are the primary determinants specifying restriction of p18m and Drt2m against Ty1c or Ty1', respectively.

## Discussion

In this study, we report highly specific retrotransposition restriction within the Ty1c and Ty1' subfamilies. Copy number control of Ty1c is mediated by the self-encoded p22/p18 restriction factor. By contrast, Ty1' seems not to produce a

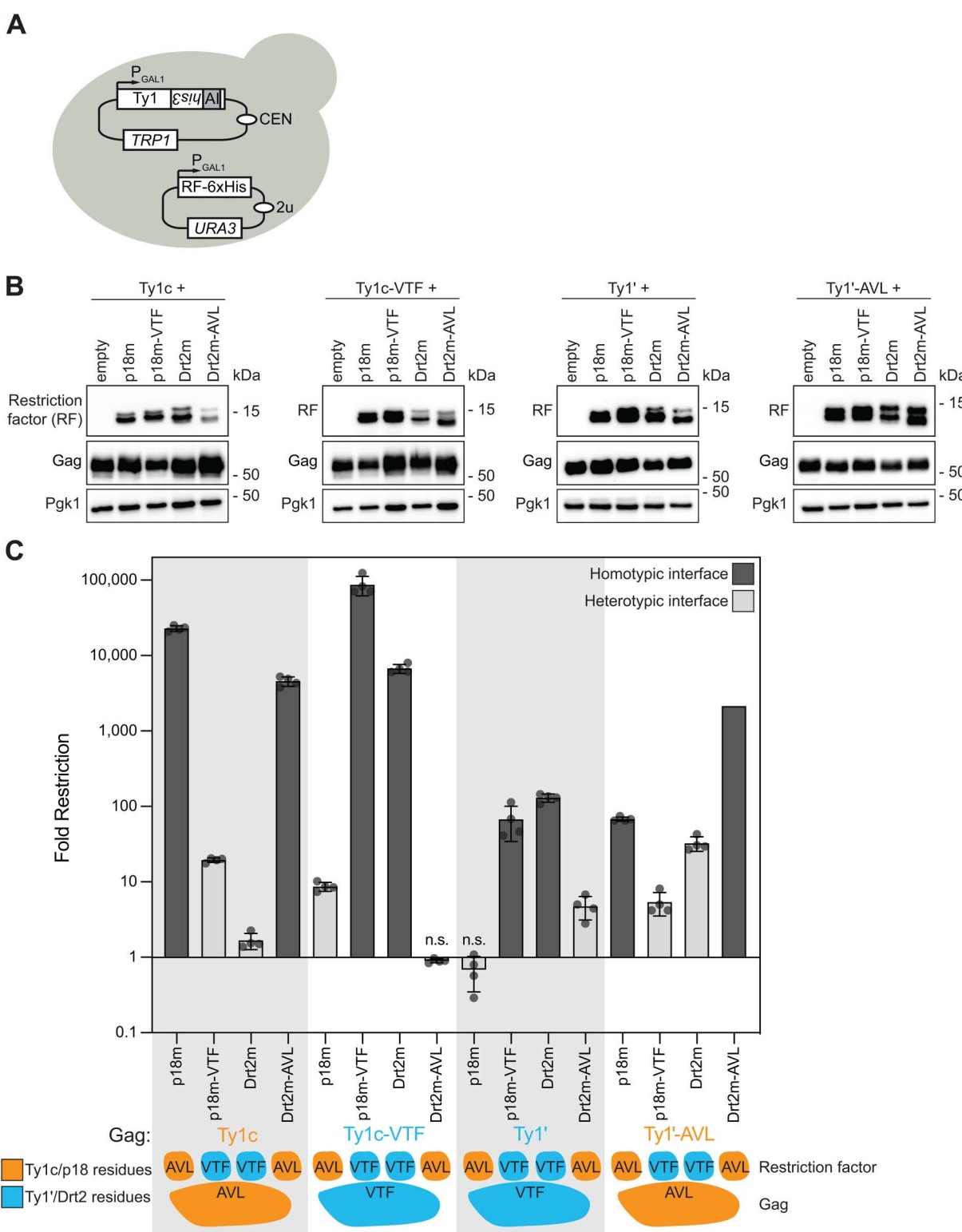

**Fig 5. Restriction factor specificity *in vivo*. (A)** Schematic illustrating the two plasmids used to co-express the transposon and restriction factor (RF). The expression of each is driven by a galactose-inducible promoter from the *GAL1* gene. The transposon is marked with the *his3-AI* retromobility indicator gene; histidine prototrophy requires retromobility. **(B)** Western blot of restriction factor and Gag expression. Protein extracts of galactose-induced

cells were immunoblotted with an anti-hexa-histidine antibody to detect restriction factors, anti-TY tag to detect Ty1c Gag, and anti-p18' to detect Ty1' Gag. Pgk1 serves as a loading control. Migration of molecular weight standards is shown alongside each immunoblot. A representative image of at least 3 replicates is shown; original images of entire-gel immunoblots are provided in the Supporting Information. **(C)** Quantitative mobility assay of galactose-induced cells. Each bar represents the mean of the four independent measurements displayed as points. The error bar center represents the mean of the four measurements and the error bar extent ± the standard deviation. Fold restriction is plotted compared to empty vector. Error bars are omitted for Ty1'-AVL with Drt2m-AVL in which no retromobility events were observed; instead, the value graphed represents the theoretical maximum fold restriction if one retromobility event had been observed. Significance is calculated from a two-sided Student's *t*-test compared with empty (n.s not significant. Exact *p*-values are provided in S1 Table). Bar color denotes homotypic or heterotypic interactions at Dimer-1. The cartoon below illustrates the Dimer-1 residues present in the restriction factor and Gag in each bar.

self-encoded restriction factor, perhaps because it lacks the appropriate sequences required for internal transcription initiation found in Ty1c *GAG*. Rather Ty1' spread is limited by Drt2, an endogenized Gag CA-CTD encoded at the fixed *DRT2* locus in the host genome. We solved the crystal structures of Drt2m and the Ty1' CA-CTD and find a highly similar structure to p18m (Fig 2). Interestingly, the Dimer-1 interface is conserved and important for the restriction mechanism in both subfamilies. The present results extend and generalize conclusions from previous work and support a common mechanism for Ty1c and Ty1' restriction that is largely due to the formation of dead-end assemblies mediated by the Dimer-1 interaction between Gag and the CA-CTD restriction factor. However, Dimer-1 appears to require complementarity at the homotypic surfaces, explaining why the Ty1' restriction factor, Drt2, does not restrict Ty1c.

Our structural data reveal how subtle differences in hydrophobic packing at an intermolecular interface affects specificity by facilitating or excluding advantageous interactions. This is most apparent at the Dimer-1 interface involving the R315-T270 interaction (Fig 2F-G). There are three residues found within Dimer-1 that diverge between the subfamilies but are highly conserved within each subfamily: AVL in Ty1c and p18 replaced by VTF in Ty1' and Drt2. The greater steric bulk of the V and F residues and the ability of T270 to hydrogen bond to a reorientated R315 create the surface differences that define interface complementarity. Thus, this arginine-flip mechanism, along with the distinct hydrophobic surfaces created by either AVL or VTF, support only the high-specificity of homotypic pairings whilst excluding incompatible heterologous pairings across the subfamilies.

Ty1 subfamily specific restriction measured in our retromobility assays (Fig 5) correlated well with exchange analyses *in vitro* (Fig 4) and VLP assembly analyses *in vivo* (Fig 6). p18m strongly restricts Ty1c *in vivo* and p18m and Ty1c-CA homodimers readily exchange subunits with one another *in vitro*. However, when the p18m Dimer-1 residues are mutated to VTF, exchange is lost, and restriction decreases by over a thousand-fold. Drt2m, which does not exchange with Ty1c-CA and very weakly restricts Ty1c retromobility, can be engineered to exchange with Ty1c-CA and robustly restrict Ty1c simply by mutating the Dimer-1 residues to AVL. Moreover, all pairwise combinations of Gag and restriction factor using both approaches corroborated; homotypic Dimer-1 interactions support exchange and restriction, and heterologous pairings do not. The effect of the subfamily specific VTF/AVL Dimer-1 sequence is remarkably strong, implicating these three residues as the primary determinants of Ty1 subfamily restriction specificity. However, mutating solely these three residues does not completely restore/abrogate restriction, suggesting there are additional, more minor contributions from other residues specifying restriction.

We observed differences in the intrinsic transposition activity and restriction magnitudes between the two subfamilies. Ty1c retromobility is an order of magnitude higher than that of Ty1', whether expressed from its own LTR-promoter or overexpressed from the *GAL1* promoter (S1 Table), indicating that Ty1c is an intrinsically more active transposon. Drt2 is less efficient at inhibiting retromobility than p22/p18. While both have tightly associating monomer-dimer $K_D$ values (Fig 3), the difference *in vivo* may be due to the role of Dimer-2, which facilitates Gag-Gag interactions as capsomeres assemble higher order VLP intermediates [30]. Ty1' and Drt2 contain an additional bulky aromatic residue at Dimer-2 (Y329) that prompted us to use further suppressing serine point mutations. *In vivo*, potentially, Ty1c Gag and p18 favor dimerization whereas Drt2 may multimerize more strongly, leading to both less efficient VLP assembly by Ty1' Gag and consequentially lower retromobility of Ty1' as well as weaker lattice poisoning restriction by Drt2.

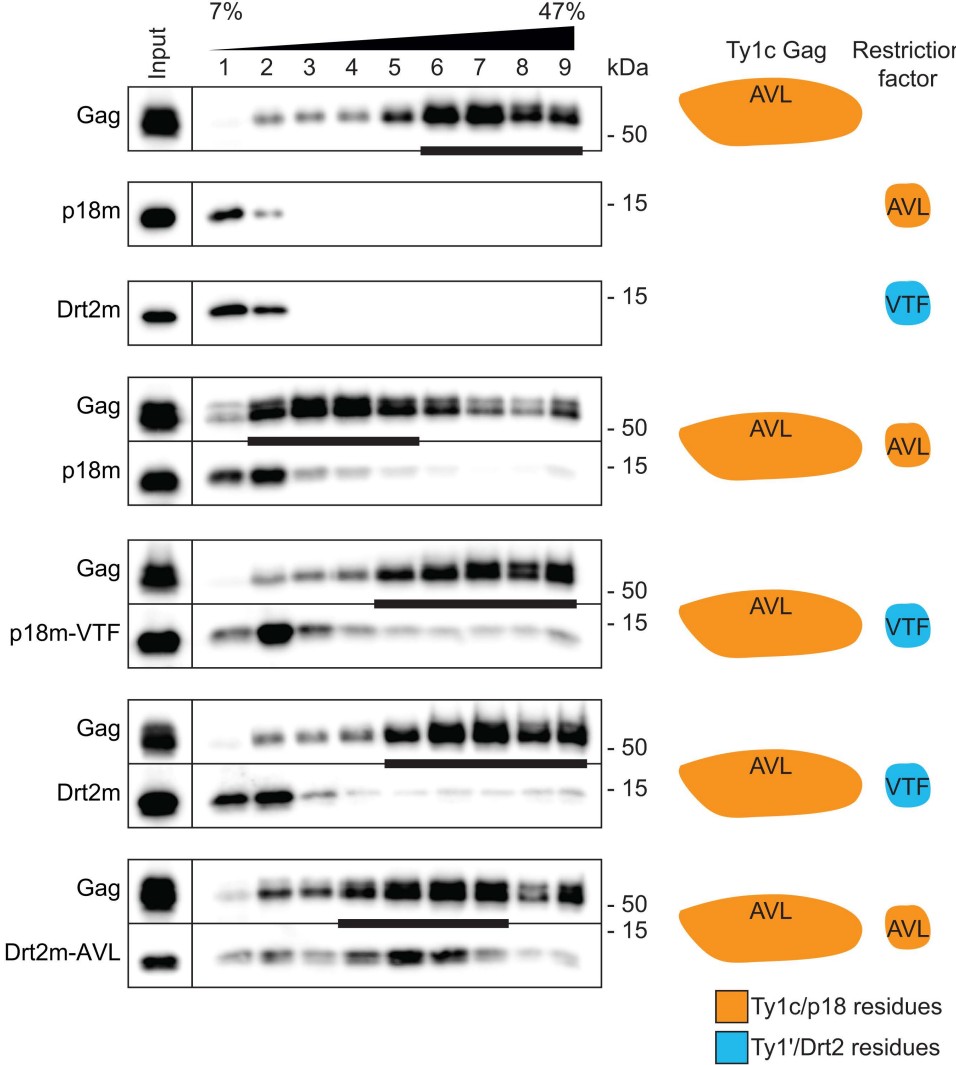

**Fig 6. Restriction factor disruption of particle assembly.** Protein extracts from galactose-induced yeast cells (Input) were fractionated over a 7-47% (w/v) continuous sucrose gradient and immunoblotted with anti-TY tag for Gag and anti-hexahistidine for restriction factors. The bars below Gag blots denote peak Gag fractions containing more than 1/9 of the Gag signal across the gradient, as determined by densitometric analysis. A representative image of at least 3 replicates is shown. Migration of molecular weight standards is shown alongside the immunoblots. Images of the whole gel immunoblots are provided in the Supporting Information. The cartoon to the right illustrates the restriction factor and Gag present in each strain with the color denoting the Dimer-1 residues of each.

Increased retrotransposition is consistent with the observation across natural yeast isolates that Ty1c is more prevalent [10,12]. The difference in Ty1c and Ty1' likely reflects their different evolutionary histories. Ty1' is the ancestral Ty1 sequence and the Ty1c subfamily diverged when a new Gag sequence was acquired via horizontal transfer from *S. paradoxus* [12]. This transfer probably included sequences permitting expression of internally initiated Ty1i RNA and p22 [32], although detailed studies of restriction of *S. paradoxus* Ty1 transposons have not been performed. Rather than acquiring three independent point mutations, the ancestral VTF in Ty1' were converted to AVL in Ty1c through the horizontal transfer event. Drt2 was likely endogenized early in the radiation of *S. cerevisiae* lineages and has been lost at least three times [34]. The *DRT2* locus is a hotbed of Ty activity and displays considerable variation across *S. cerevisiae* lineages, including

many LTR remnants from Ty1–4, full-length Ty2 and full-length and truncated Ty1 elements. Our restriction specificity data suggest that the highly active Ty1c transposon would not have been inhibited by pre-existing Drt2 and Ty1c could therefore spread and overtake Ty1' as the more abundant Ty1 element in many lineages of *S. cerevisiae*. p22-mediated CNC increases restriction potency only with additional genomic copies of Ty1c present because p22 is produced from the self-encoded Ty1i transcript [31,32,38]. The fact that the mechanisms of restriction in both subfamilies involve lattice poisoning via a CA-CTD emphasizes not only that VLP assembly is an essential part of the life cycle of Ty1-family retro-transposons, but that CA interactions are a potent target for evolution of restriction.

Given that many yeast strains harbor multiple families of Ty transposons in a single genome, it will be interesting to investigate additional combinations of Ty elements for interactions across families [10]. While Ty1 family transposons are the most abundant in many lab strains, Ty2 is more prevalent amongst natural isolates and there has been introgression of Ty2 sequence into Ty1 subfamilies [10]. Whether there is cross-restriction between Ty2 and certain Ty1 subfamilies or between other Ty elements remains to be determined. Furthermore, the convergence of Ty1c and Ty1' restriction mechanisms raise the possibility that other CA-CTD, or even CA-NTD, restriction factors have not been identified for other transposons, whether exapted or self-encoded. Given the highly specific restriction conferred by the respective CA-CTDs restriction factors, it may be possible to engineer a synthetic CA-CTD to target the capsid of a transposon or virus with no identified restriction factor, extending previous work addressing whether synthetic fusion proteins containing well-characterized domains or segments can restrict HIV and MLV replication [39]. Further, investigating the consequences of introducing internal Gag promoters, equivalent to that leading to expression of the p22 protein of Ty1c, into other Ty transposons might also reveal more examples of copy number control by lattice poisoning.

Considering the pronounced evolutionary and functional relationships between LTR-retrotransposons and retroviruses, mechanisms of host defense also evolved that target Gag or CA [40]. Murine Fv1 provides an important paradigm for retroelement restriction and viral tropism as Fv1 alleles encode retroelement CA proteins that inhibit certain murine leukemia retroviruses (MLV) post-entry, after reverse transcription but prior to integration [35]. Two major alleles have been described, Fv1$^n$ and Fv1$^b$, which restrict B-tropic MLV and N-tropic MLV, respectively. Constitutive expression of Fv1 is low and, like Ty1c p22/p18, restriction can be saturated by high virus titers or Ty1 overexpression [32,33,41]. Fv1 restriction targets CA, and analogous to Ty1/p22 and Ty1'/Drt2, amino acid substitutions clustered in CA can alter restriction specificity [42–45]. Moreover, although the structural basis of Fv1 restriction specificity remains incompletely defined, it likely results from the differential avidity of Fv1$^n$ and Fv1$^b$ oligomer-binding to the assembled MLV capsid lattice [46,47]. Similarly, the specificity of the CA-binding by the HIV restriction factor Trim5α is largely determined by Trim5α PRY/SPRY domain interactions with CA target residues, with further contributions from the CA-binding cis-trans isoprolyl-isomerase CypA [48,49].

Phylogenomic analyses of the human genome reveal five separate Gag domestication events of the *Metaviridae* family comprising at least 24 genes [50]. Furthermore, a growing number of Gag-like genes participate in normal cellular processes [51]. The exapted Gag-like proteins RTL8 and PEG10 are family members. Like p22 and Drt2, RTL8 alone does not form VLPs as it lacks a full-length capsid. PEG10 is required for embryonic development, forms VLPs containing packaged RNA, and has been engineered as an RNA delivery vehicle [52]. Recent work suggests that RTL8 Gag inhibits PEG10 VLP function by co-assembling with PEG10 to form mixed VLPs [53]. The RTL8-PEG10 interaction decreases the level of PEG10 VLPs and increases the pool of unassembled PEG10, suggesting the defect may occur early in assembly. These results raise the possibility that RTL8 is a natural regulator of PEG10 VLP function utilizing an assembly-blocking mechanism akin to p22 and Drt2 restriction of Ty1 VLP assembly and Gag and Gag-Pol maturation [30,32]. Moreover, in LINE-1 (L1) non-LTR retrotransposons that comprise 17% of the human genome [54,55], one domesticated derivative of L1 ORF1p, L1TD1, enhances L1 retrotransposition [56]. Contrastingly, another subset of naturally occurring L1 elements encode truncated ORF1 proteins that inhibit retrotransposition [57]. Thus, in both human L1 and yeast Ty1, retrotransposon domestication has resulted in exapted genes that modulate the process of retrotransposition.

In conclusion, we have utilized synergistic biochemical, genetic, and structural approaches to show that Ty1 copy number control and restriction is mediated by Dimer-1 compatibility and that restriction is subfamily-specific. Our work reinforces how a common restriction mechanism for Ty1c and Ty1' evolved from independent trajectories as p22/p18 is self-encoded by Ty1c and was horizontally transferred into *S. cerevisiae*, while Drt2 is encoded by a nonfunctional element, widely distributed and present in ancient lineages [12,30,34]. A common restriction mechanism affecting Ty1 VLP assembly through the action of a conserved CA domain and parallels with regulation by mammalian exapted genes suggests that similar ways of modulating retroelement movement or repurposed CA-CA interactions may exist in other organisms and that could be developed therapeutically.

## Methods

### Yeast strains, plasmids, and media

Strains and plasmids used in this study are listed in S4 and S5 Tables, respectively. Standard yeast genetic and microbiological techniques [58] were used throughout this work. All Ty1 canonical nucleotide and amino acid information corresponds to Ty1H3 sequence (GenBank M18706.1) [59]. Ty1' nucleotide and amino acid information corresponds to YBLWTy1-1 (SGD ID: S000006808) and Drt2 information corresponds to the *DRT2* locus of strain UWOPS05-227.2 [34]. Amino acid residues for Gag proteins and p18m or Drt2m refer to the coordinates in Gag. p18m and Drt2m constructs expressed in yeast span residues M249-R355 and include a C-terminal PLEHHHHHH tag [30,34]. Plasmid cloning was performed by inserting custom commercial gene fragments (Integrated DNA Technologies and Twist Bioscience) with NEBuilder HiFi DNA Assembly Master Mix (New England Biosciences cat. no. E2621) between the XhoI and EcoRI sites of pBDG1293, BbvCI and BstEII sites of pBDG1534, or BbvCI and NruI sites of pBDG1697. All plasmids generated were verified by DNA sequencing. For galactose induction in liquid media, starter cultures were grown overnight at 30 °C in synthetic media containing 2% raffinose, diluted 1:20 into media containing 2% galactose, and grown at 22 °C as previously described [30].

### Ty1his3-AI mobility

Ty1 retromobility events were detected using the *his3-AI* retromobility indicator gene [37] by quantitative assays [32,34]. Quantitative retromobility frequencies were determined from quadruplicate cultures diluted in water, plated on synthetic dropout media, and colonies counted. All experiments were galactose-induced for 48-hours at 22 °C, except for strains DG4259 and DG4303–5 which were grown in glucose media. Data represent at least four independent inductions and is representative of at least three separate experiments; *p*-values were calculated by two-sided Student's *t*-test. Complete data, including standard deviations and *p*-values, are listed in S1 Table.

### Amino acid sequence analysis

Conservation of p18m, Drt2m, and Ty1' amino acid sequences was visualized using WebLogo v2.8.2 [60]. For p18m, a total of 98 sequences were analyzed. 97 *S. cerevisiae* p18m amino acid sequences, Gag residues 259–355, were extracted from full-length Ty1c elements [12]. p18m from Ty1H3 (GenBank M18706) was also included in the analysis. For Drt2m, the region homologous to p18m was extracted from 15 *S. cerevisiae* strains containing *DRT2* [34]. For Ty1' Gag, the region homologous to p18m was extracted from 35 *S. cerevisiae* Ty1' amino acid sequences [12]. Amino acid percent identity calculations were performed using Clustal Omega [61] and reported as the percentage of amino acids that are 100% identical across all the compared sequences.

### Immunoblotting

Immunoblotting of total protein extracted from galactose-induced yeast by trichloroacetic acid (TCA) precipitation was analyzed using standard techniques [32]. Cells were broken by vortexing in the presence of glass beads in 20% TCA

and washed in 5% TCA. Proteins were separated on 15% (for detecting p18m and Drt2m constructs) or 10% (for detecting Pgk1 and Gag) SDS-PAGE gels. PVDF membranes were immunoblotted with antibodies at the following dilutions in 2.5% milk-TBST: monoclonal mouse hexa-histidine antibody clone HIS.H8 (ThermoFisher cat. no. MA1–21315) (1:3000) for His-tagged restriction factors, mouse monoclonal anti-TY tag antibody clone BB2 (1:5000) [62] for Ty1 Gag, rabbit polyclonal anti-p18' (Boster Bio cat. No. DZ33974) (1:1000) for Ty1' Gag, or mouse monoclonal anti-Pgk1 antibody clone 22C5D8 (Invitrogen cat. no. 459250) (1:1000) for Pgk1 loading control. Immune complexes were detected with Western-Bright enhanced chemiluminescence (ECL) detection reagent (Advansta cat. no. K-12049-D50). Imaging was performed using a ChemiDoc MP (Bio-Rad). Precision Plus Kaleidoscope protein standards (Bio-Rad cat. no. 1610395) were used to estimate molecular weights.

### Sucrose gradient sedimentation

Following 48-hour galactose induction, a 100 mL culture was harvested, and cells were broken in 15 mM KCl, 10 mM HEPES- KOH, pH 7, 5 mM EDTA containing RNase inhibitor (100 U per mL), and protease inhibitors (16 µg ml$^{-1}$ aprotinin, leupeptin, pepstatin A and 2 mM PMSF) in the presence of glass beads. Cell debris was removed by centrifugation at 10,000 x g for 10 min at 4°C. Approximately five milligrams total protein in 500 µL buffer was applied to a 7–47% (w/v) continuous sucrose gradient and centrifuged using a SW41 Ti rotor at 25,000 rpm (77,000 x g) for 3 hr at 4°C. After centrifugation, 9 x 1.2 mL fractions were collected and normalized volumes of input and fractions were immunoblotted with anti-TY or anti-p18' antibodies to detect Gag and hexa-histidine antibody to detect restriction factors [33]. Densitometric analysis was performed using Image Lab (Bio-Rad).

### Protein expression and purification

The DNA sequences for Drt2m (residues M259 to R355), Ty1' CA-CTD (residues M259-Q351) and Ty1c CA (residues V169-N355) were synthesized, codon-optimized for expression in *E. coli*, by GeneArt (Thermo Fisher Scientific) (S6 Table). The genes were amplified by PCR and inserted into a pET22b expression vector (Novagen) between the NdeI and XhoI restriction sites to produce C-terminal fusion proteins containing the hexa-histidine tag PLEHHHHHH. The p18m construct has been previously described [30]. Mutations to suppress Dimer-2 interactions (F323S, Y326S and Y329S) and the M259L in Ty1c CA were introduced into parent constructs using the Quikchange II XL site directed mutagenesis kit (Agilent) following the manufacturer's instructions. The primer sequences for PCR and mutagenesis are provided in S7 Table. The mutant constructs to convert the Dimer-1 interfaces (V266A, T270V and F312L; Drt2m to p18m) and (A266V, V270T and L312F; p18m to Drt2m) were produced by gene synthesis and inserted into pET22b by GeneArt (S6 Table).

p18m WT and mutant proteins were expressed and purified as described [30]. Drt2m, Ty1' CA-CTD and Ty1c CA proteins were expressed in the *E. coli* strain BL21 (DE3) grown in LB- or TB-media by induction of log phase cultures (OD$_{600}$ = 0.8) with 1 mM IPTG, followed by incubation overnight at 19 °C with shaking. Cells were pelleted by centrifugation, washed with PBS and then resuspended in 50 mM Tris-HCl pH 8.5, 150 mM NaCl, 10 mM Imidazole, 3 mM MgCl$_2$ supplemented with 1 mg mL$^{-1}$ lysozyme (Sigma-Aldrich), 10 µg mL$^{-1}$ DNase I (Sigma-Aldrich) and 1 Protease Inhibitor cocktail tablet (EDTA free, Pierce) per 40 mL of buffer. For Drt2m and Ty1' CA-CTD cells were lysed using an EmulsiFlex-C5 homogenizer (Avestin) and His-tagged protein captured from the clarified lysate using immobilized metal ion affinity on a 5 mL Ni$^{2+}$-NTA Superflow column (Qiagen). For Ty1c CA, the protein was found in pellet fraction after lysate clarification and was recovered into the supernatant by extraction with high salt buffer (50 mM Tris-HCl pH 8.5, 1 M NaCl, 10 mM Imidazole, 1m M TCEP), after stirring for 1 hour at 4 °C followed by further clarification. All proteins were applied to Ni$^{2+}$-NTA Superflow columns equilibrated in 50 mM Tris-HCl pH 8.5, 150 mM NaCl, 25 mM Imidazole, 1mM Tris (2-carboxyethyl) phosphine (TCEP) then washed with at least 20 column volumes of the equilibration buffer. Proteins were eluted with 50 mM Tris-HCl pH 8.5, 150 mM NaCl, 250 mM Imidazole. Carboxypeptidase A (Sigma C9268) was added at 1:100 (w:w)

ratio and the resulting mixture incubated overnight at 4 °C to digest the C-terminal his-tag. The Carboxypeptidase A was then inactivated by the addition of 5 mM TCEP and proteins further purified by gel filtration chromatography on a Superdex 75 (26/60) column equilibrated in 20 mM Tris-HCl, 150 mM NaCl, 1 mM TCEP pH 8.5 for Drt2m and Ty1' CA-CTD and in 20 mM Tris-HCl, 1M NaCl, 1 mM TCEP pH 8.5 for TY1c CA.

Electrospray-ionization mass spectrometry (ESI-MS) was used to determine protein molecular masses of mutants and confirm His-tag removal where appropriate. Usually complete digestion left a C-terminal PLEH or PLE remnant. Purified restriction factor and CA proteins were concentrated to 1–7 mM or 0.2 mM respectively by centrifugal ultrafiltration (Vivaspin, MWCO 10 kDa), then snap frozen and stored at -80 °C. Protein concentrations were determined by UV absorbance spectroscopy using extinction coefficients calculated at 280 nm derived from the tryptophan and tyrosine content.

## Protein crystallization

All proteins were crystallized by sitting drop vapor diffusion at 18 °C, using Swissci MRC 2-drop trays (Molecular Dimensions). 200 nL drops, with a ratio of 3:1 protein to mother liquor were set using a Mosquito robot with incorporated humidity chamber (TTP Labtech). Ty1' CA-CTD crystals were obtained from conditions containing 12 mg/mL Ty1' CA-CTD in gel filtration buffer incubated with 0.2 M NaCl, 10% PEG 3000, 0.1 M sodium phosphate/citrate pH 4.2. Small shard-like crystals ~60x30 µm appeared within 1–2 days. Ty1' CA-CTD (F323S) crystals were obtained from conditions containing 70 mg/mL Ty1' CA-CTD (F323S) in gel filtration buffer incubated with 0.2 M NH$_4$Cl, 20% PEG 6000, 0.1 M Tris-HCl pH 8.0. Large ~200 µm 3-dimensional prismoid crystals appeared after 100–202 days. Drt2m(SS) crystals were obtained from conditions containing 50 mg/mL Drt2m(SS) in gel filtration buffer incubated with 0.2 M NaF, 20% PEG 3350, 0.1 M Bis-Tris-Propane pH 7.5. Rhomboid crystals with dimensions ~50–70 µm appeared between 3–7 months. Prior to data collection, all crystals were harvested by looping into drops containing mother liquor supplemented with 25% PEG 200 cryoprotectant and flash frozen in liquid nitrogen.

## Data collection and structure determination

X-ray diffraction data were collected at the Diamond Light Source (DLS) beamlines, I03, I04 and I24 (Didcot, UK). Grid-scanning was required to ensure that only well-diffracting portions of crystals was exposed to the beam. Data were processed using the Xia2 pipeline [63], using DIALS [64] and AIMLESS [65] or with anisotropic scaling applied using Autoproc/STARANISO (http://staraniso.globalphasing.org/cgibin/staraniso.cgi). The structures were solved by molecular replacement in PHASER [66] implemented in the CCP4 interface [67], using a polyalanine model of p18m (from PDB: 7NLH) as an initial search model for the Ty1' CA-CTD and Ty1' CA-CTD (F323S) structures and then the Ty1' CA-CTD (F323S) model was used to solve the Drt2m(SS) structure. For Ty1' CA-CTD after initial rebuilding using ArpWarp [68] and manual building in COOT 0.9.6 [69], the structure was refined using REFMAC5 [70]. For Ty1' CA-CTD (F323S) the structure was rebuilt with Buccaneer [71] and refined using PHENIX [72]. For Drt2m(SS), the Ty1' CA-CTD (F323S) structure was first used to guide rebuilding in COOT 0.9.6 that produced a good model for Chain A and B. Subsequently Chain B was used as a reference model for restrained refinement of the C and D chains in PHENIX. TLS groups determined using TLSMD [73] were included in the final rounds of refinement when models were near complete. Throughout refinement, model geometries were monitored and assessed using Molprobity [74] and PDB-REDO [75]. Details of data collection and structure refinement statistics are presented in S2 Table.

## Structure analysis and sequence alignment

Electron density maps were visualized in COOT 0.9.6 and PDB files from crystal structures were viewed and rendered in PyMOL 2.5.5 (Schrodinger, LLC). Analysis of protein interfaces within crystal structures was performed with PDB-PISA [76]. The DALI server [77,78] was used to obtain comparative 3D alignments of p18m, Ty1' CA-CTD and Drt2m. Primary sequences were aligned in Megalign-Pro using Clustal-W default settings and implemented within the DNAStar software suite.

## SEC-MALLS

Size exclusion chromatography coupled multi-angle laser light scattering (SEC-MALLS) was used to determine the molar mass distribution of p18m variants, Drt2m and Ty1c CA derivatives as well as combined p18m - Ty1c CA pairs. Individual proteins as well as equimolar mixtures (50 µM) of p18m - Ty1c CA pairs were first incubated at 277 K for 18–20 hours. Samples (100 µL) were then applied to a Superdex INCREASE 200 10/300 GL column equilibrated at 298 K in high salt buffer: 50 mM Tris-HCl, 800 mM NaCl, 0.5 mM TCEP, 3 mM NaN$_3$ pH 8.5 at flow rate of 1.0 mL min$^{-1}$. The scattered light intensity and the protein concentration of the column eluate were recorded using a DAWN-HELEOS laser photometer and OPTILAB-rEX differential refractometer respectively. The weight-averaged molecular mass of material contained in chromatographic peaks was determined from the combined data from both detectors using the ASTRA software version 7.3.2.19 (Wyatt Technology Corp., Santa Barbara, CA, USA).

## Exchange data analysis

To assess the degree of exchange between p18m, Drt2m and variants with Ty1c CA and Ty1c CA-VTF, data from the SEC-MALLS analysis was combined with quantitation of the intermediate exchange peak magnitude that was observed when p18m, Drt2m, Ty1c CA, Ty1c CA-VTF mixtures were resolved by SEC. Briefly, from the recorded differential refractive index measurements a residual signal at each time interval (t) across the peak elution envelope ($S_{res(t)}$), was determined by subtraction of the signal in the calculated sum chromatogram of the individual p18m and Ty1c CA components $S^{p18}_{calc(t)} + S^{CA}_{calc(t)}$ from that in the experimentally observed p18m - Ty1c CA chromatogram ($S^{p18+CA}_{Exp(t)}$), equation (1).

$$S_{res(t)} = S^{p18+CA}_{Exp(t)} - \left( S^{p18}_{calc(t)} + S^{CA}_{calc(t)} \right)$$

(1)

The fraction sum residual ($\theta_{res}$) was then calculated from equation (2) after integration of $|S_{res(t)}|$ and $|S^{p18}_{calc(t)} + S^{CA}_{calc(t)}|$ over the entire peak elution envelope using integration limits ($_{i, j}$) of 14–18 minutes using Profit 7.1.5 software (Quantum Soft, Switzerland).

$$\theta_{res} = \frac{\int_i^j |S_{res(t)}|}{2 \int_i^j \left( |S^{p18}_{calc(t)} + S^{CA}_{calc(t)}| \right)}$$

(2)

Restriction factor – CA combinations with ($\theta_{res}$) greater than 2x the average background value derived from all non-exchanging pairs (2.2 ± 0.3%) were scored as exchanging.

## Analytical ultracentrifugation

Sedimentation equilibrium experiments were performed in a Beckman OPTIMA-AUC analytical ultracentrifuge using conventional aluminum double sector centerpieces and sapphire windows in an An50-Ti rotor maintained at 293 K. Solvent density and the protein partial specific volumes were determined as described [79]. Protein refractive index increments were calculated based as described [80]. Prior to centrifugation, p18m and Drt2m were prepared by exhaustive dialysis against the buffer blank solution (50 mM Tris-HCl, 150 mM NaCl, 0.5 mM TCEP, pH 8.5). Samples (150 µL) and buffer blanks (170 µL) under-layered with 20 µL and 10 µL of Fluorochem FC-43 (Merck), respectively, were loaded into the cells and after centrifugation for 30 hours, interference and UV absorbance data (280 nm) were collected at 2 hourly intervals, using the OPTIMA-AUC Experiment Portal Software. Over this period no changes in sample concentration distribution profiles were observed. The rotor speed was then increased, and the procedure repeated. Data were collected at three speeds 20,000 rpm (29,120 x $g$), 23,000 rpm (38,511 x $g$) and 26,000 rpm (49,213 x $g$) on samples at different concentrations ranging from 20 to 90 µM. The program SEDPHAT [81] was used to initially determine weight-averaged molecular

masses by nonlinear fitting of individual multi-speed equilibrium profiles to a single-species ideal solution model. Inspection of these data revealed that the molecular mass showed significant concentration dependency and gave poor fits to a single species model. Therefore, global fitting of the data to a monomer-dimer-tetramer model incorporating the data from both detection systems, multiple speeds and multiple sample concentrations was applied to extract monomer-dimer ($K_D^{1-2}$) and dimer-tetramer ($K_D^{2-4}$) equilibrium dissociation constants.

## Supporting information

**S1 Fig. Amino acid conservation of *S. cerevisiae* Ty1c p18m, Drt2m, and Ty1' Gag.** Amino acid coordinates and alpha helix positions (green bars) are indicated above the alignment. Three subfamily-specific residues are highlighted at positions 266, 270, and 312 (yellow boxes). Y-axis indicates sequence conservation measured in bits.
(PNG)

**S2 Fig. Expression and activity of serine solubility mutants. (A)** Western blot of restriction factor and Gag expression. Protein extracts of galactose-induced cells were immunoblotted with an anti-hexa-histidine antibody to detect restriction factors and anti-p18' to detect Ty1' Gag. Pgk1 serves as a loading control. Migration of molecular weight standards is shown alongside the immunoblot. A representative image of at least 3 replicates is shown, original images of entire-gel immunoblots are provided in the Supporting Information. **(B-C)** (*Upper*) Quantitative mobility assay of galactose-induced cells. Each bar represents the mean of the four independent measurements displayed as points. The error bar center represents the mean of the four measurements and the error bar extent ± the standard deviation. Significance is calculated from a two-sided Student's *t*-test compared with wildtype (n.s not significant, *** $p < 0.001$. Exact *p*-values are provided in S1 Table). In **B**, fold restriction is plotted compared to empty vector and fold-change in restriction compared to wildtype is indicated above the bars. In **C**, retromobility frequency in the absence of restriction factors is plotted. (*Lower*) Schematic illustrating the separate plasmids used to express the transposon and restriction factor (RF). The expression of each is driven by a galactose-inducible promoter from the *GAL1* gene. The transposon is marked with the *his3-AI* retromobility indicator gene; histidine prototrophy requires retromobility.
(PNG)

**S3 Fig. Crystal structure asymmetric units of Ty1' CA-CTD and Drt2m. (A)** The asymmetric units of the Ty1' CA-CTD, Ty1' CA-CTD (F323S) and Drt2m(SS) crystal structures. Protomers in each asymmetric unit are shown in cartoon representation colored wheat (M1), green (M2), pale blue (M3) and pink (M4). The equivalent Dimer-1 dimer interfaces are indicated with the arrows. **(B)** 3D structural superposition of Ty1' CA-CTD (F323S) and Drt2m(SS) dimers. Backbone representations are colored green and pale blue respectively. Structures were aligned using 160 backbone Cα atoms, yielding an RMSD of 0.5 Å. Equivalent α-helices are labelled sequentially from N- to C-terminus. **(C)** Sequence alignment of Ty1' Gag residues 259–355 and Drt2m. Numbering is according to the equivalent position in Ty1' Gag. Positions of α-helices observed in crystal structures are indicated by the green bars above the alignment. Divergent residues are indicated by boxes, thick red boxes indicate Dimer-1 interface residues V, T, F where Ty1' Gag and Drt2m are identical. Gray shaded residues indicate residues where Dimer-2 suppression serine mutations have been made.
(PNG)

**S4 Fig. SEC-MALLS analysis of Ty1c-CA, Drt2m and p18m restriction factors. (A)** Ty1c CA(F323S), **(B)** Ty1c CA-VTF(F323S), **(C)** p18m(F323S), **(D)** Drt2m(SSS), **(E)** p18m-VTF(F323S) and **(F)** Drt2m-AVL(SSS). In each panel dRI is plotted against retention time, the molar mass, determined at 1-second intervals throughout peak elution, is plotted as points and the sample loading concentrations are indicated; 200 μM (red), 100 μM (orange), 50 μM (green), 25 μM (blue) and 12.5 μM (violet). Monomer and dimer molar masses are indicated with the dashed lines.
(PNG)

**S5 Fig. Exchange peak residual analysis of restriction factor specificity.** Dimer subunit exchange of (**A-D**) Ty1c CA(F323S) and (**E-H**) Ty1c CA-VTF(F323S) with p18m(F323S), p18m-VTF(F323S), Drt2m(SSS) and Drt2m-AVL(SSS) restriction factors are shown. In each panel, the experimentally measured dRI chromatogram (Exp) for 50 μM equimolar loadings of Ty1c CA(F323S) or Ty1c CA-VTF(F323S) with p18m(F323S), p18m-VTF(F323S), Drt2m(SSS) and Drt2m-AVL(SSS) restriction factors is shown in blue. The calculated sum chromatogram (Sum) derived by addition of individual 50 μM loading chromatograms for each pair is shown in red. The residual chromatogram (Residual) obtained after subtraction of the Sum chromatogram from the Exp chromatogram is shown in black. The orange dashed line indicates the baseline of zero residual. Integration over the peak elution envelope of the Exp, Sum and Residual chromatograms and application of equation-1 and equation-2 (see methods) was used to assess the degree of Ty1c CA(F323S) and Ty1c CA-VTF(F323S) subunit exchange with each p18m or Drt2m restriction factor.
(PNG)

**S1 Table. Retromobility frequencies.**
(PDF)

**S2 Table. X-ray data collection and structure refinement statistics.**
(PDF)

**S3 Table. Hydrodynamic parameters and sedimentation equilibrium data.**
(PDF)

**S4 Table. Yeast strains used in this study.**
(PDF)

**S5 Table. Yeast plasmids used in this study.**
(PDF)

**S6 Table. Codon-optimized gene sequences.**
(PDF)

**S7 Table. Cloning and mutagenesis primers.**
(PDF)

**S1 Raw Images. Raw images of immunoblots corresponding to 5B, 6 and S2A Figs.**
(PDF)

**S1 Data File. Source numerical data for Figs 1, 5, S2, S3, and S4 and sequence data for S1.**
(XLSX)

## Author contributions

**Conceptualization:** Sean L Beckwith, Matthew A Cottee, J. Adam Hannon-Hatfield, Jonathan P Stoye, Ian A Taylor, David J Garfinkel.

**Formal analysis:** Sean L Beckwith, Matthew A Cottee, J. Adam Hannon-Hatfield, Abigail C Newman, Emma C Walker, Justin R Romero, Ian A Taylor.

**Funding acquisition:** Sean L Beckwith, Ian A Taylor, David J Garfinkel.

**Investigation:** Sean L Beckwith, Matthew A Cottee, J. Adam Hannon-Hatfield, Abigail C Newman, Emma C Walker, Justin R Romero, Ian A Taylor.

**Supervision:** Sean L Beckwith, Jonathan P Stoye, Ian A Taylor, David J Garfinkel.

**Validation:** Sean L Beckwith, Matthew A Cottee, J. Adam Hannon-Hatfield.

**Visualization:** Sean L Beckwith, Matthew A Cottee, Ian A Taylor.

**Writing – original draft:** Sean L Beckwith, Ian A Taylor, David J Garfinkel.

**Writing – review & editing:** Sean L Beckwith, Matthew A Cottee, J. Adam Hannon-Hatfield, Jonathan P Stoye, Ian A Taylor, David J Garfinkel.

## Acknowledgments

We gratefully acknowledge the Diamond Light Source, Didcot, UK (Grant No MX25587) and beamlines I04, I03, and I24 for access. We also acknowledge the Crick Structural Biology and Proteomics Technology Platforms for access and expertise.

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
