## [Decision Letter · Decision Letter 0]

1 Sep 2025

PGENETICS-D-25-00848

Probing the molecular determinants of Ty1 retrotransposon restriction specificity in yeast

PLOS Genetics

Dear Dr. Taylor,

Thank you for submitting your manuscript to PLOS Genetics. The manuscript was reviewed by three experts in the field.  All three found the study sound, interesting, and exciting.  Each had suggestions for revisions to address points raised by review.  Therefore, we invite you to submit a revised version of the manuscript that addresses the points raised during the review process.  The three reviewers each indicated minor revisions as their recommendation.  Their comments seem straightforward to address, but if you have any questions, feel free to be in touch.

Please submit your revised manuscript within 30 days Oct 01 2025 11:59PM. If you will need more time than this to complete your revisions, please reply to this message or contact the journal office at plosgenetics@plos.org. Please include the following items when submitting your revised manuscript:

We look forward to receiving your revised manuscript.  We appreciate your entrusting this very nice study to PLOS Genetics!

Best wishes,

Joe

Joseph Heitman, MD, PhD

Academic Editor

PLOS Genetics

Geraldine Butler

Section Editor

PLOS Genetics

Aimée Dudley

Editor-in-Chief

PLOS Genetics

Anne Goriely

Editor-in-Chief

PLOS Genetics

**Journal Requirements:**

1) Please provide an Author Summary. This should appear in your manuscript between the Abstract (if applicable) and the Introduction, and should be 150-200 words long. The aim should be to make your findings accessible to a wide audience that includes both scientists and non-scientists. Sample summaries can be found on our website under Submission Guidelines:

https://journals.plos.org/plosgenetics/s/submission-guidelines#loc-parts-of-a-submission

- TM on pages: 24, and 27.

5) Thank you for stating "The atomic coordinates and structure factors for Ty1' CA(M259-Q351), Ty1' CA(M259-Q351) (F323S), and Drt2m(SS) have been deposited in the Protein Data Bank under accession numbers 9RXW, 9RXX, and 9RXY, respectively." Please note that, though access restrictions are acceptable now, your entire minimal dataset will need to be made freely accessible if your manuscript is accepted for publication. This policy applies to all data except where public deposition would breach compliance with the protocol approved by your research ethics board.

6) Please ensure that the funders and grant numbers match between the Financial Disclosure field and the Funding Information tab in your submission form. Note that the funders must be provided in the same order in both places as well. Currently,  the order of the funders is not exactly the same in both places.

7) Thank you for indicating that "The authors declare no competing financial or non-financial interests." 

Please revise your current Competing Interest statement to the standard "The authors have declared that no competing interests exist."

**Reviewers' comments:**

Reviewer's Responses to Questions

Reviewer #1: In this study, authors provide robust evidence to elucidate the mechanism of restriction specificity between two closely-related retrotransposons within the Ty1 family (Ty1c and Ty1’). Using a genetic approach, they demonstrate that restriction specificity is largely dependent on three residues in the C-terminal capsid (CA-CTD) domain of p18/p22 and Drt2 restriction factors which differ (AVL vs VTF residues, respectively). Further, authors determine crystal structures for Drt2m and Ty1’ CA revealing the similarity between the minimal restriction factors (p18m and Drt2m) and demonstrate that determinant residues are located on the conserved Dimer-1 hydrophobic interface with Ty1 Gag capsid. The most compelling evidence for the proposed mechanism is that in vivo swapping of AVL and VTF residues for each restriction factor changes the restriction specificity for Ty1c and Ty1’, respectively.

Though the individual mechanisms describing Ty1c and Ty1’ restriction were largely worked out previously, this study demonstrates that restriction is specific for each element and additionally provides new mechanistic insights as to how differences in just a few residues in two highly conserved CA-CTD regions of analogous factors can confer remarkable restriction specificity and thus specific ‘lattice-poisoning control’ of a particular Ty1 element. Authors posit that such specificity has application for designing inhibitors that precisely target CA-CTD domains of viruses and transposons to suppress mobility.

The study is well-written, employs detailed and rigorous methodology, and combined multiple biophysical and genetic approaches to corroborate the proposed mechanism. Very elegant and convincing work that will be of broad interest to readers studying retrotransposon/retroviral control as well as those studying protein structure-function relationships.

Comments/questions:

The introduction provides a useful and detailed primer on the mechanisms of Ty1c and Ty1’ restriction mechanisms necessary for understanding the proposed study. As a reader, I was also seeking a brief introduction of literature with analogous restriction systems beyond the Ty elements in S. cerevisiae to better understand the context for the study and potential for broader implications. In the discussion, authors highlight the Fv1 restriction system for murine leukemia virus which is described as analogous to the Ty1/p22 and Ty1’Drt2 system, where AA substitutions clustered in the capsid can also alter restriction specificity. I think a nod to this body of literature in the introduction, instead of the discussion alone, is appropriate and further provides a basis for the rationale of the approach.

Similarly, I was curious about other Ty elements (Ty2, Ty4, Ty5) and Ty3 families. Lines 379-318 in the discussion state “whether there is cross-restriction between Ty2 and certain Ty1 subfamilies or between other Ty elements remains to be determined.” Has the mechanism of restriction for these elements been determined or relevant proteins/domains defined that can be compared to Ty1 restriction? A more detailed explanation would be helpful.

Does the S. cerevisiae background strain used for in vivo experiments contain Ty1c or Ty1’ genomic copies? If so, would this influence the Ty1c and Ty1’ reporter-based mobility/restriction (for example p22/p18 production from genomic copies)?

For the plasmids with reporter constructs and restriction factors introduced, is the approximate copy number per cell known and whether plasmids are stably maintained? A brief mention of this would be helpful, though the consistency in the fold-restriction data across experiments suggests stable maintenance.

Minor suggested edits:

The following terms may not be familiar to most readers – suggest briefly defining when first introduced:

Line 30, 82: ‘endogenized’ restriction factor

Line 72: ‘sub-genomic’

Line 120: include Gag in this sentence for clarity: “and the p18m region of Ty1' Gag is 92.8%...”

Fig 1 E and F – is there a reason for the dotted lines outlining the images? Suggest no outline or solid outline instead.

Fig 5C: Error bars appear to be missing on final Drt2m-AVL lane.

Reviewer #2: Probing the molecular determinants 1 of Ty1 retrotransposon restriction specificity in yeast

Sean Beckwith et al

This manuscript is a continuation of work from the Garfinkel lab to understand the molecular details of mechanisms that restrict the copy number of LTR-retrotransposons in Saccharomyces cerevisiae. Their previous work identified a self-limiting mechanism of Ty1 which expresses a segment of capsid (p18) that inhibits transposition activity. Structural characterization of p18 identified a dimer interface, dimer-1, that appears to compromise particle assembly. Other work from the Garfinkel lab identified a subfamily of the Ty1 family, Ty1’ which unlike Ty1, is suppressed by a domesticated restriction factor DRT2.

The current manuscript demonstrates that the inhibitory actions of p18 and DRT2 are highly specific for Ty1 and Ty1’, respectively. Both restriction factors exhibit structures highly similar to the C-terminal fragments of Ty1 and Ty1’ capsid (CA). Sequence analysis of 148 Ty1’ and 98 Ty1 (p18) elements identified three invariant residues in the dimer-1 interface with A266, V270, and L312 encoded by p18 and V266, T270, and F312 in both Ty1’ and DRT2. Structures of Ty1’ and DRT2 revealed that these three divergent residues are at the center of the hydrophobic dimer-1 interface and that they adapt an altered conformation that is incompatible with the three divergent residues in Ty1 and p18. The authors tested the possibility that the structural differences at these three dimer-1 residues are responsible for the specificity of the restriction factors. Through measures of protein association and retrotransposition frequencies the authors find that the three divergent residues are necessary and sufficient for the restriction specificity. Importantly, swapping the three residues in p18 and DRT2 resulted in changing the restriction specificity to the opposite retrotransposon. In addition, the authors demonstrate that the restriction activities of p18 and DRT2 result from disrupting the assembly of virus-like particles.

The quality of the data is high, and the manuscript is well written. The genetic and structural results provide a detailed understanding of the Ty1 restriction mechanisms. This level of detail is missing in most if not all examples of retrovirus restriction factors such as Fv1 that are domesticated fragments of Gag. The details of Ty1 restriction are thus a valuable model for understanding how assembly of Gag-like particles is altered. The importance of understanding this process is highlighted recently in Science (PMID: 40773553).

Detailed comments:

1. Line 58, I would list proteins in order of their position in Pol, PR, IN, and RT.

2. Line 62, Is it clear that reverse transcription of Ty1 occurs in the cytoplasm. This view has changed with HIV-1 with strong evidence reverse transcription occurs in the nucleus.

3. Line 99, what is the relationship between the presence of DRT2 and Ty1' across populations of cerevisiae? Do strains lacking Ty1' have DRT2?

4. Line 118, Fig. S1 title indicates that Ty1c is also shown. But it’s not. I understand that p18m is the same sequence as Ty1c so why not label it with both names. p18m and Ty1c.

5. Line 137 and throughout. It would be best to call this CA-CTD. You do list the segment M259-Q351, but it would be best to indicate its actually the C-terminal half of CA.

6. Line 172, Fig 2D and E appear to be of the monomer not the dimer. I would like to see these residues in a dimer making clear it’s the interface in an image like Fig. 2A. In such a drawing you can highlight the dimer interactions of the 3 key residues. It would be easier to understand and would complement the blowup shown in Figs 2F and G.

7. Line 175, It would help to ref. Fig. 2 F and G here. Helix alpha 3 is not labeled in Fig 2F or 2G.

8. Line 217, Please present a general concept of SEC-MALLS.

9. Line 246, Reference Table 1 here.

10. Line 250, These exchange factors show a clear trend at the higher levels. But the exchanges are not obvious in Fig. 4 for the lower levels such as the difference between 6.2% of Drt2m-AVL with CA. vs the 2.3% of Drt2m-AVL with CA-VTF. The former represents exchange while the latter no exchange. Are there statistics that can be applied that can distinguish the exchangers from the non-exchangers?

11. Line 280, Although most of the differences in Fig 5 are obviously significant, there is one pair that could use a statistical test. Ty1'-AVL comparing p18m to Drt2M. Does p18m show a statistically higher fold restriction than the Drt2m?

12. Line 309, If DRT2 is fixed, is Ty1' older than Ty1? What is the context of the Drt2 gene. Is there a remnant of the LTR, does it have other recognizable parts of Ty1. Does the promoter of DRT2 share regulatory sequence with Ty1' that is independent from Ty1 expression? An interesting evolutionary question is why did p18 evolve to be self-limiting while DRT2 is domesticated?

13. Line 397, should reference PMID: 40773553.

Reviewer #3: In this work, Beckwith et al. investigate the molecular basis of Ty1 retrotransposition restriction in S. cerevisiae. Previously, they demonstrated that retrotransposition of the Ty1c family, predominant in the laboratory strain S288C, is repressed by the self-encoded restriction factor p22 (and its maturation product p18), which is identical to GAG-CTD. This restriction occurs via hydrophobic interactions between p22/p18 and GAG-CTD termed the dimer-1 interface, preventing Virus-Like Particle (VLP) formation, which is an essential step in Ty1 retrotransposition. In an independent study, they found that retrotransposition of a Ty1 subfamily (Ty1’), under-represented in S288C but prevalent in other Saccharomyces strains, is restricted via a similar mechanism involving this time an endogenized DRT2 gene that encodes a Ty1’ GAG-CTD equivalent to p22.

Combining genetic, structural and biophysics approaches, here they reveal strong similarities in the crystal structure of the minimal restriction factors p18m and Drt2m and more specifically in their dimer-1 interfaces. However, despite these similarities, each restriction factor remains specific to its Ty1 family. This specificity depends mostly on three amino acids present in the dimer-1 interface, which are not conserved between the two restriction factors and create slightly different intermolecular interfaces. By swapping the three amino acids between the two restriction factors and/or Ty1c and Ty1’ sequences they convincingly demonstrate that the three amino acids (AVL/VTF) are sufficient to trigger dimer formation and restriction specificity. In fact, Ty1 restriction is only possible when the combination of three amino acids is identical in the restriction factor and the Gag protein expressed by Ty1.

Overall, I found this article to be solid and interesting. The experiments are clear and the data corroborate the conclusions drawn. Perhaps the results are not entirely unexpected, but the fact that three amino acids are necessary and sufficient to ensure restriction specificity while maintaining a similar 3D structure between the two restriction factors is nonetheless remarkable. As the authors point out in their discussion and argue with examples in the literature, this supports the idea that such a restriction mechanism may have emerged repeatedly to “self-regulate” other retroelements.

Comments.

Figure 1 and the text in the results section that refers to it.

The text lacks details to understand the retromobility assay. First, retromobility is a term specific to Ty retrotransposons and should be explained. A brief explanation of the his3-AI system would also help those unfamiliar with it. Second, it should be clarified in the text (at least in the legend of Figure 1) that the assay is performed in a Ty1c-less context to avoid Ty1c restriction by endogenous Ty1c elements.

The restriction of Ty1’ by DRT2 is quite weak (Fig 1C). This could mean that Drt2 is limiting in fully restricting Ty1’. What happens when DRT2 is overexpressed?

Given that the retromobility of Ty1c is several orders of magnitude higher than that of Ty1’, we cannot exclude from these data that Drt2 levels are limiting in restricting Ty1c. Even if the subsequent data indicate that this is mainly due to sequence specificity, this hypothesis should at least be mentioned at this stage of the manuscript.

It could also be relevant to show data comparing the RNA and Gag levels of Ty1c and Ty1’.

Fig 1C and Table S1: x107 should be x10-7

Fig 1D and the corresponding section “Subfamily specific residues in the Dimer-1 interface”, you should refer to the materials and methods section, which describes in more detail which sequences were used, otherwise the paragraph seems a bit vague.

Figure S1, and Tables S4 and S5: It is not easy to understand which strains have been used in the retromobility assays. Could you add a column in Table S4 to indicate in which experiments the strains have been used?

Figure 2 and the text in the results section that refers to it.

Line 120: p18m should be Drt2-like? In the section “Subfamily specific residues in the Dimer-1 interface”, you should refer to the materials and methods section, which describes in more detail which sequences were used, otherwise it's difficult to follow.

Fig S2B and S2C: Ty1 should be Ty1c in the graphs and the drawings.

Fig S2C: x107 should be x10-7

Figure 4 and Table 1. Drt2m(SSS) seems to make week exchanges not only with Ty1-CA(F323S) VTF but also Ty1-CA(F323S). Could you discuss this point?

Figure 5 and Figure 6.

These are very nice data.

Fig 5B. The Western blots seem to indicate that Drt2m-AVL is more abundant/stable in the presence of Ty1'-AVL. Is this observation reproducible? Do you have a hypothesis based on the other results?

Minor comments:

Line 49: “cerevisiae” must be in italic

Line 51: Why ref. 7?

Line 100: “… in diverse strains”, Please, add “but absent in classical laboratory strains”.

Line 110-114. The sentence needs to be clarified.

Line 120: p18m should be p18m-like or Drt2-like?

Line 146: Please, refer to Fig S2B.

Line 178: “of the greater steric bulk”, remove one “of the”.

Line 224: Please, refer to Fig 1E and 1F.

**Have all data underlying the figures and results presented in the manuscript been provided?**

Reviewer #1: Yes

Reviewer #2: Yes

Reviewer #3: Yes

PLOS authors have the option to publish the peer review history of their article (what does this mean? ). If published, this will include your full peer review and any attached files.

**Do you want your identity to be public for this peer review?** For information about this choice, including consent withdrawal, please see our Privacy Policy .

Reviewer #1: No

Reviewer #2: **Yes: ** Henry Levin

Reviewer #3: No

**Figure resubmission:**

**Reproducibility:**



---

## [Editor Report · Decision Letter 1]

29 Sep 2025

Dear Dr Taylor,

We are pleased to inform you that your revised manuscript entitled "Probing the molecular determinants of Ty1 retrotransposon restriction specificity in yeast" has been editorially accepted for publication in PLOS Genetics. Congratulations!

Yours sincerely,

Joe

Joseph Heitman, MD, PhD

Academic Editor

PLOS Genetics

Geraldine Butler

Section Editor

PLOS Genetics

Aimée Dudley

Editor-in-Chief

PLOS Genetics

Anne Goriely

Editor-in-Chief

PLOS Genetics

BlueSky: @plos.bsky.social

Comments from the reviewers (if applicable):

**Data Deposition**

http://datadryad.org/submit?journalID=pgenetics&manu=PGENETICS-D-25-00848R1

**Press Queries**

---

## [Editor Report · Acceptance letter]

PGENETICS-D-25-00848R1

Probing the molecular determinants of Ty1 retrotransposon restriction specificity in yeast

Dear Dr Taylor,

We are pleased to inform you that your manuscript entitled "Probing the molecular determinants of Ty1 retrotransposon restriction specificity in yeast" has been formally accepted for publication in PLOS Genetics! Your manuscript is now with our production department and you will be notified of the publication date in due course.

With kind regards,

Zsofia Freund

PLOS Genetics

On behalf of:
